# Monocyte progenitors give rise to multinucleated giant cells

Anne Kathrin Lösslein [1,2], Florens Lohrmann[1,3,4,17], Lisa Scheuermann[5,17], Kourosh Gharun [1,6], Jana Neuber[1,6], Julia Kolter[1], Aaron James Forde[1,6], Christian Kleimeyer[1], Ying Yee Poh[7], Matthias Mack[8], Antigoni Triantafyllopoulou [1,9], Micah D. Dunlap [10,11], Shabaana A. Khader[10,11], Maximilian Seidl [12], Alexandra Hölscher[13], Christoph Hölscher[13,14], Xue Li Guan[7], Anca Dorhoi [5,15,16] & Philipp Henneke [1,3 ✉]

The immune response to mycobacteria is characterized by granuloma formation, which features multinucleated giant cells as a unique macrophage type. We previously found that multinucleated giant cells result from Toll-like receptor-induced DNA damage and cell autonomous cell cycle modifications. However, the giant cell progenitor identity remained unclear. Here, we show that the giant cell-forming potential is a particular trait of monocyte progenitors. Common monocyte progenitors potently produce cytokines in response to mycobacteria and their immune-active molecules. In addition, common monocyte progenitors accumulate cholesterol and lipids, which are prerequisites for giant cell transformation. Inducible monocyte progenitors are so far undescribed circulating common monocyte progenitor descendants with high giant cell-forming potential. Monocyte progenitors are induced in mycobacterial infections and localize to granulomas. Accordingly, they exhibit important immunological functions in mycobacterial infections. Moreover, their signature trait of high cholesterol metabolism may be piggy-backed by mycobacteria to create a permissive niche.

[1] Institute for Immunodeficiency, Center for Chronic Immunodeficiency, Medical Center and Faculty of Medicine, University of Freiburg, Freiburg, Germany. [2] MOTI-VATE Graduate School, Faculty of Medicine, University of Freiburg, Freiburg, Germany. [3] Center for Pediatrics and Adolescent Medicine, Medical Center and Faculty of Medicine, University of Freiburg, Freiburg, Germany. [4] Spemann Graduate School for Biology and Medicine (SGBM) and IMM-PACT Clinician Scientist Program, Faculty of Medicine, University of Freiburg, Freiburg, Germany. [5] Max Planck Institute for Infection Biology, Berlin, Germany. [6] Faculty of Biology, University of Freiburg, Freiburg, Germany. [7] Nanyang Technological University, Lee Kong Chian School of Medicine, Singapore, Singapore. [8] University Hospital Regensburg, Internal Medicine II, Nephrology, Regensburg, Germany. [9] German Rheumatism Research Centre Berlin, Leibniz Association, Berlin, Germany. [10] Department of Molecular Microbiology, Washington University in St. Louis School of Medicine, Saint Louis, MO, USA. [11] Department of Pathology and Immunology, Washington University in St. Louis School of Medicine, Saint Louis, MO, USA. [12] Center for Chronic Immunodeficiency and Institute for Clinical Pathology, Department of Pathology, Medical Center and Faculty of Medicine, Freiburg, Germany and Institute of Pathology, Heinrich Heine University and University Hospital of Duesseldorf, Duesseldorf, Germany. [13] Forschungszentrum Borstel, Leibniz Lungenzentrum, Borstel, Germany. [14] Deutsches Zentrum für Infektionsforschung, Standort Borstel, Borstel, Germany. [15] Institute of Immunology, Federal Research Institute for Animal Health, Friedrich-Loeffler-Institut (FLI), Insel Riems, Germany. [16] Faculty of Mathematics and Natural Sciences, University of Greifswald, Greifswald, Germany. [17] These authors contributed equally: Florens Lohrmann, Lisa Scheuermann. ✉email: Philipp.henneke@uniklinik-freiburg.de

The development of the *Homo sapiens sapiens* has been in coevolution with mycobacteria. It can be safely assumed that this process is ongoing, since even in highly developed countries, humans are exposed to mycobacteria on a daily basis. Environmental mycobacteria with potential virulence for humans such as *Mycobacterium avium* are regularly found in drinking waters[1,2]. Moreover, an estimated 2 billion people worldwide are infected with *M. tuberculosis* (*M.tb*)[3] and most of these infections are latent, i.e. asymptomatic[4]. In other words, tight immunological control is essential to prevent the individual from succumbing to mycobacterial infection, even after decades of coexistence[5]. A well-appreciated site of mycobacterial latency is the unique multicellular granuloma reaction[6]. Distinct types of macrophages (MΦ), particularly multinuclear giant cells (MGC), contribute to the core of granulomas. Although the first description of MGC in tuberculous granuloma was published 150 years ago[7] and their origin from leukocytes has long been postulated[8], mechanisms underlying mycobacterial MGC formation have still been incompletely resolved. We have recently shown that the development of polyploid MGC involves cell-autonomous affliction of DNA damage and impairment of p53 function by the potent antimycobacterial effector nitric oxide (NO), resulting in mitotic defects and multinucleation[9,10]. Thus, much enlarged MΦ that avidly phagocytose mycobacteria and infected apoptotic cells arise[9,10]. However, mycobacteria may override the antibacterial activity of MGC, turning them into dedicated mycobacterial niches within the human host[10]. Despite the striking cell biological properties of MGC, their origin remains poorly understood. Dissection of the MGC pedigree seems particularly important, regarding the recent revolution in understanding MΦ development. In particular, it remains to be resolved, if MGC are the progeny of a specific cell lineage or subset, and what metabolic preconditions might determine the differentiation path towards an MGC. Mononuclear phagocytes, i.e., monocytes, MΦ, and dendritic cells (DC) share a common progenitor, the monocyte-MΦ-DC progenitor (MDP)[11,12], although recent data rather indicate that MDP and GMP constitute independent progenitors of monocytes[13]. The position of MDP in definitive hematopoiesis is based on experimental data from mice and receives further support by the recent identification of the human MDP analogue[14]. An even more committed clonogenic common monocyte progenitor (cMoP), which renews monocytes and MΦ independently of DC, has been identified in the bone marrow and spleen of mice[15], as well as in human bone marrow and umbilical cord blood[16]. Whereas the role of cMoP to monocyte and MΦ development is well established, biological functions beyond myeloid cell development have not been assigned to cMoP, yet.

Here, we introduce cMoP and its progeny, the circulating inducible monocyte progenitor (iMoP) as immune cells with a particular capacity for MGC transformation. The high cholesterol and lipid metabolism in cMoP is an essential prerequisite for MGC formation and may favor intracellular mycobacterial survival, since cholesterol serves as an energy source for *M.tb*[17]. Together, we provide firm evidence for a so far unappreciated role of monocyte progenitors in mycobacterial infections. They inextricably integrate progenitor characteristics, such as high proliferative activity and low apoptosis, with specific functions in mycobacterial immunity.

## Results

### cMoP are dedicated progenitors for multinucleated giant cells.
We previously found that MΦ progenitors have a higher potential than mature MΦ to undergo transformation into MGC upon infection with mycobacteria or TLR2 stimulation (Fig. 1a

and Herrtwich et al.)[9]. However, since the progenitor characteristics remained undefined, we decided to systematically explore the identity of the MGC progenitor. Thus, we stringently sorted monocytes (MC), cMoP, MDP, and Lin⁻ c-kit⁺ progenitors from the bone marrow of *Cx3cr1$^{gfp/+}$* reporter mice (Fig. 1b). In agreement with the most widely accepted definitions (Auffray et al.)[11], (Hettinger et al.)[15], we identified MDP as CD11b⁻ CD117⁺ CD115⁺ CX3CR1⁺ Ly6C⁻. cMoP differed from MDP with respect to Ly6C expression (CD11b⁻ CD117⁺ CD115⁺ CX3CR1⁺ Ly6C⁺)[15]. MC were defined as CD11b⁺ CD117⁻ CD115⁺ CX3CR1⁺, and Lin⁻ c-kit⁺ progenitors as CD11b⁻ CD117⁺ CD115⁻ (Fig. 1c). The primary and most robust endpoint was the formation of MGC, which were defined as cells with at least three nuclei and an increased cytoplasmic area (Fig. 1a). Using this definition, cMoP exhibited an exceedingly high ability to form MGC in response to the mycobacterial glycolipid lipomannan (LM). Multinucleation and cell surface enlargement was evident already after 4 days, and increased further until day 6 (Figs. 1d and e). In contrast to cMoP, the quantitative MGC formation was only marginally above base line in MC and Lin⁻ c-kit⁺ cells under the same conditions (Fig. 1e). The recently disputed cMoP progenitor MDP[18] showed some, albeit delayed and overall reduced capacity to form MGC, compared to cMoP (Fig. 1e). The high MGC-forming potential of cMoP was confirmed by in vitro infection with the *M.tb*-complex species *M. bovis* Bacillus Calmette-Guérin (BCG) (Fig. 1f and Supplementary Figs. 1a and 1b). Under all these conditions, cMoP were not only potent, but also quite specific MGC progenitors, since besides absolute quantity, the relative proportion of cMoP-derived MGC was also increased compared to MC (Figs. 1d, g and Supplementary Fig. 1c). In extension of our relatively stringent definition of MGC as cells with at least 3 nuclei, we wondered if binucleation may serve as a marker for the transformation process into MGC. Indeed, we observed an absolute predominance of binucleated cells in cMoP compared to MC after exposure to BCG (Supplementary Fig. 1c). Yet, the total number of cells with two nuclei was highest in cMoP treated with M-CSF alone (Supplementary Fig. 1d). This was likely due to the complex balance of binucleated cells as an indicator of high multiplication rates, which resolves upon further proliferation (M-CSF control), and binucleation as an indicator of mitotic defects promoting polyploidy and multinucleation (BCG stimulation). More immature progenitors as MDP responded to stimulation with mycobacterial glycolipids predominantly with proliferation into mononuclear daughter cells (Supplementary Fig. 1e). Similar to monocytes from the bone marrow (MC), blood monocytes did not show MGC-forming properties, when stimulated with LM (Fig. 1g and Supplementary Fig. 1f).

### cMoP combine macrophage and progenitor characteristics.
In full agreement with their progenitor status and similar to previously published data[15], cMoP showed rapid proliferation with total CFSE dilution already after 3 days in vitro (Fig. 2a). Stimulation with LM markedly decreased proliferation over the entire course of the experiment (Fig. 2b). These observations were confirmed by analysis of the cell density after stimulation with fixed *M. bovis* BCG (Fig. 2c). Next, we determined, whether cMoP differed from MC in their cell death response to mycobacteria. We found increased apoptosis in MC compared to cMoP, when stimulated with fixed *M.tb* (Fig. 2d). In line with these observations, MC stimulated with BCG showed a decreased mitochondrial potential compared to cMoP, which is indicative of apoptosis (Supplementary Fig. 1g). Accordingly, viability of cMoP exposed to mycobacteria and their glycolipids exceeded

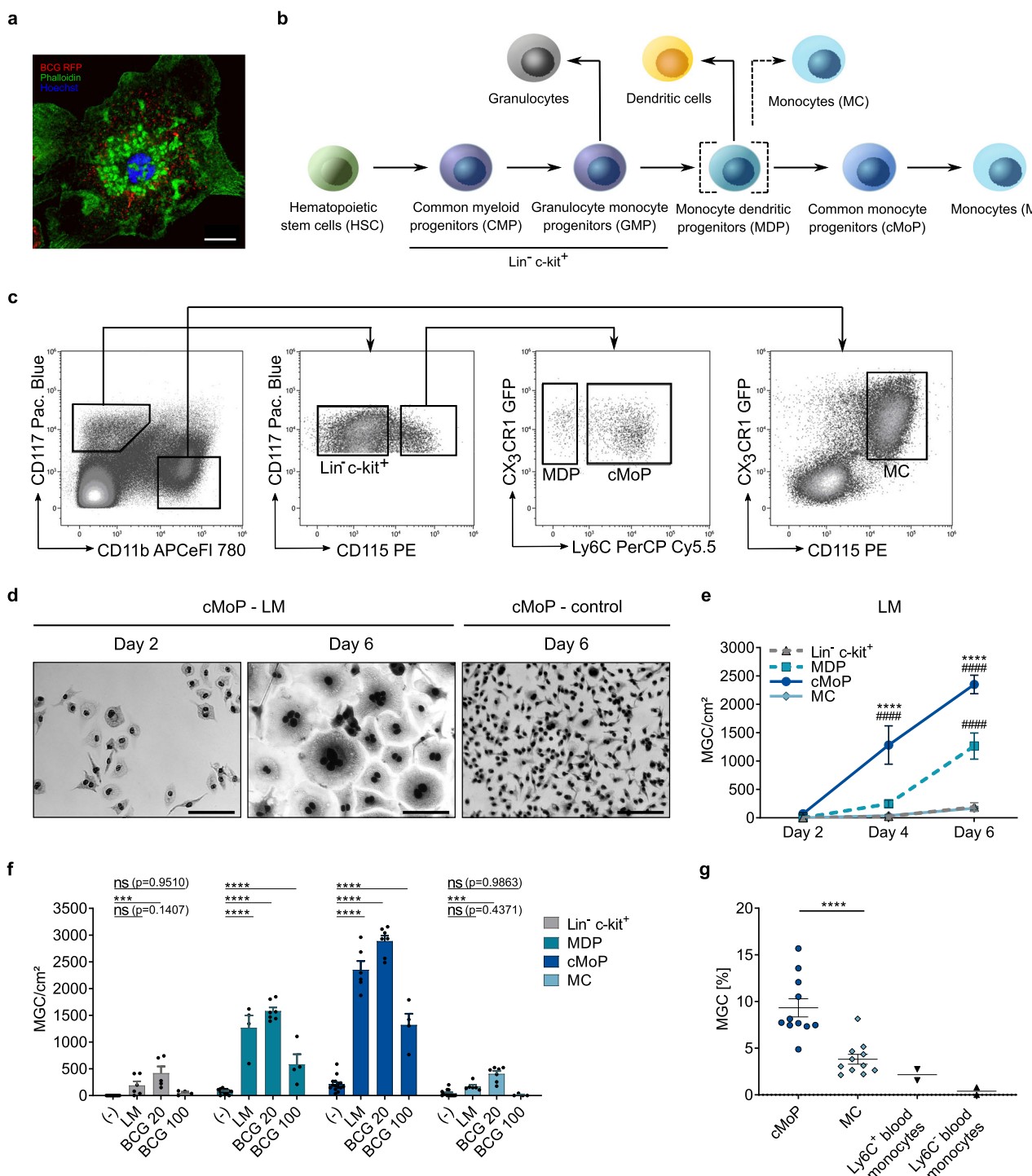

that of MC. Since cMoP were highly activated by the Toll-like receptor (TLR) 2 agonist LM (Fig. 1e), we wondered whether cMoP differed from other myeloid cell subsets in regard to TLR expression. Notably, we found cMoP to express TLR2 and TLR6, the cognate receptors of diacylated glycolipids[19]. In contrast, the expression of TLR13, the receptor for bacterial RNA, was very low in cMoP, yet high in MC (Fig. 2e). Since MDP and MC showed similar or higher expression of TLR2 and TLR6 than cMoP, we concluded that TLR expression was probably not a limiting factor in MGC formation by MC (Fig. 2e). These observations, together with those on survival and the superior MGC-forming properties of cMoP in the presence of myco-bacteria, spurred the hypothesis that cMoP may be important in mycobacterial immunity. Thus, we measured induction of TNFα formation and nitric oxide (NO) as established components of mycobacterial immunity and found cMoP to be highly active for both parameters (Figs. 2f and g). High NO activity was of par-ticular interest, since we previously found it to be one of the drivers of MGC formation via induction of DNA damage and impairment of p53[10]. In full accordance with a central role of NO in the mycobacterial response of cMoP, LM and BCG-induced MGC formation was abrogated in cMoP from inducible nitric oxide synthase-deficient ($inos^{-/-}$) mice (Fig. 2h). Altogether, cMoP showed characteristic progenitor traits of high pro-liferative activity combined with signature properties for myco-bacterial immunity.

**Fig. 1 cMoP are dedicated progenitors for multinucleated giant cells. a** Phalloidin (green) and Hoechst (blue) staining of a cMoP-derived MGC after stimulation with BCG-RFP (red). Scale bar: 20 μm. Channels were adjusted individually with respect to contrast and brightness. **b** Scheme of monocyte/macrophage cell development in murine bone marrow (BM). **c** FACS gating for sorting of cMoP, MDP, Lin⁻ c-kit⁺ cells and MC from bone marrow of *Cx3cr1ᵍᶠᵖ/+* mice. **d** Representative Hemacolor stainings of cMoP from *Cx3cr1ᵍᶠᵖ/+* mice, stimulated with LM (Lipomannan from *M. smegmatis*, 4 μg/ml) for 2 or 6 days compared to M-CSF (50 ng/ml) controls. Scale bar: 100 μm. Gamma-correction was adjusted to 0.8 in all images. **e** cMoP (dark blue), MDP (cyan, dashed), Lin⁻ c-kit⁺ cells (gray, dashed) and MC (light blue) were stimulated with LM (4 μg/ml) for 2, 4, and 6 days. Quantification of MGC per cm² based on Hemacolor staining (compare Fig. 1d). Depicted are mean ± SEM of *n* = 3 (MDP D2 and D4), *n* = 4 (MDP D6), *n* = 4 (cMoP, MC, Lin⁻ c-kit⁺ D2 and D4) and *n* = 6 (cMoP, MC, Lin⁻ c-kit⁺ D6) biologically independent samples. ****p < 0.0001 (cMoP – MDP), ####p < 0.0001 (refers to MC and Lin⁻ c-kit⁺); (two-way ANOVA, Tukey's multiple comparisons test (MCT)). **f** cMoP (dark blue), MDP (cyan), Lin⁻ c-kit⁺ cells (gray) and MC (light blue) from *Cx3cr1ᵍᶠᵖ/+* mice were stimulated with LM (4 μg/ml) or BCG (MOI 20, MOI 100) for 6 days and quantified for MGC formation by Hemacolor staining. Bars show mean ± SEM of *n* = 13–15 (control), *n* = 4–6 (LM), *n* = 5–7 (BCG MOI 20) and *n* = 4 (BCG MOI 100) biologically independent samples examined over 3–9 experiments. Detailed numbers of samples (*n*) for each condition are presented in Supplementary Table 3. ns = not significant, ***p = 0.0003, ****p < 0.0001 (two-way ANOVA, Dunnett's MCT). **g** Ly6C⁺ and Ly6C⁻ blood monocytes (black) and BM-derived cMoP (dark blue) and MC (light blue) were stimulated with LM (4 μg/ml) for 6 days. MGC formation was quantified based on Hemacolor staining. Graph shows mean ± SEM of *n* = 11 biologically independent samples for cMoP and MC, and *n* = 2 biologically independent samples for blood monocytes. ****p < 0.0001 (student's unpaired *t*-test, two-tailed).

## cMoP-derived MΦ keep their potential to form MGC and show a characteristic metabolic profile.

cMoP are an intermediate state between the multipotent progenitors for mononuclear phagocytes and monocytes. In view of the distinct immune cell functions of cMoP, we analyzed the transformation kinetics of cMoP in culture. We found the cMoP immunophenotype to be very transient, since already after 4 h in vitro almost two thirds of cMoP were negative for the progenitor marker CD117. In contrast, CD11b expression increased more slowly, with ~50% of cMoP expressing CD11b at 12 h, and thus acquired an immunophenotype reminiscent of monocytes (CD11b⁺ Ly6C⁺ CD117⁻), (Fig. 3a and Supplementary Fig. 2a). Stimulation with LM did not impact on the loss of CD117 expression, yet resulted in faster upregulation of CD11b (Supplementary Fig. 2b). After 48 h in culture with M-CSF, cMoP could no longer be distinguished in flow cytometry from monocyte-derived macrophages with respect to the typical MΦ-surface markers CD11b and F4/80 (Fig. 3b).

These findings resulted in the conundrum that cMoP, which rapidly acquired a monocyte immunophenotype in vitro, but not MC, were capable of transforming into MGC. Thus, we asked, whether cMoP retained the MGC-forming capability, even when the expression of identity-defining surface markers had turned into that of inflammatory monocytes. To address this question, we cultured cMoP in M-CSF for 48 h, which resulted in an early macrophage phenotype as outlined above (Fig. 3b), and then added LM for 4 days. Subsequently, we determined MGC formation and compared this with MGC generated from cMoP, which were stimulated with LM for 4 days directly after sorting. Intriguingly, both preparations did not quantitatively differ in giant cell formation (Fig. 3c). These observations indicated that surface marker expression alone was not sufficient to predict the MGC-forming potential of MΦ species. Therefore, we analyzed the transcriptional program of cMoP, MC, and their respective descendants after 48 h of in vitro differentiation. We cultured cMoP and MC in M-CSF only, or additionally with LM for 48 h, a time point when both cell types became very similar with respect to the expression of CD117, CD11b, and Ly6C. Notably, after 48 h most of the cells originating from cMoP or MC expressed F4/80, thereby qualifying as in vitro generated MΦ (Fig. 3b). Transcriptomic analysis revealed a multitude of differentially regulated genes between cMoP and MC after LM stimulation (Supplementary Figs. 2c and 2d). In particular, MC down-regulated genes of key enzymes for cholesterol and fatty acid metabolism upon TLR2 activation (Fig. 3d). In contrast, the respective lipid metabolism genes were not downregulated when cMoP were stimulated in a similar fashion, indicating that metabolic changes were connected with MGC transformation.

Yet, the expression of the dehydrocholesterol reductase 24, which catalyzes the production of cholesterol from desmosterol[20], as well as HMG-CoA-reductase, the rate-limiting enzyme for cholesterol synthesis and squalene epoxidase[21], were upregulated in LM-stimulated cMoP compared to LM-stimulated MC (Figs. 3d and e). Corresponding to this, *Msmo1* and *Lss*, two genes involved in steroid metabolism and terpenoid backbone biosynthesis[22], were upregulated in LM-stimulated cMoP (Fig. 3d). The same regulatory mechanisms were observed for key enzymes of fatty acid synthesis, such as fatty acid synthase and fatty acid desaturase (Figs. 3d and e). In contrast, the carnitine o-palmitoyltransferase 1, which mediates β-oxidation[23] and therefore degradation of fatty acids, was transcriptionally downregulated in activated cMoP compared to MC. In addition, lipid binding molecules and transporters were differentially regulated in cMoP and MC (Fig. 3d and Supplementary Fig. 2e). Pathway analysis also revealed an upregulation of cholesterol biosynthesis in cMoP stimulated with LM compared to similarly treated MC. In contrast, the latter showed an upregulation of apoptosis signaling, toll receptor signaling, and chemokine/cytokine-mediated inflammatory signaling (Fig. 3f). Together, these findings linked the regulation of lipid metabolism to MGC formation, and in particular to the transformation potential of cMoP.

Cholesterol is an integral component of cell membranes and cholesterol synthesis is therefore essential in single-cell growth and proliferation[20]. High cholesterol metabolism in cMoP corresponded to their strong proliferative activity (Fig. 2a), as determined by the CFSE assay. In line with this, we found upregulation of the proliferation marker *Mki-67* in cMoP with, and without LM stimulation, compared to MC (Supplementary Fig. 2f).

Further differences in lipid biosynthesis between cMoP and MC concerned prostaglandins, which are well-established intermediates in the regulation of acute and chronic inflammation[24]. Whereas *Ptges*, *Ptgs1* and *Ptgs2*, which encode for enzymes in prostaglandin H2 and E2 biosynthesis, were upregulated in stimulated MC as compared to cMoP, *Hpgds*, which is involved in prostaglandin D synthesis, was increased in cMoP (Fig. 3d). Differences in the expression of inflammatory effector molecules are further signs of discrete immune regulator properties of these cells. In LM-stimulated MC, genes encoding for interleukins like IL-1β and chemokines like Cxcl3, Cxcl2, and Ccl5 (Supplementary Figs. 2g and h) were upregulated. In cMoP, chemokines such as Ccr5, Ccl8, and Ccl12 were potently induced by LM (Supplementary Fig. 2h). Since chemokine and interleukin production are essential for the immune response to mycobacteria[25], the contrast

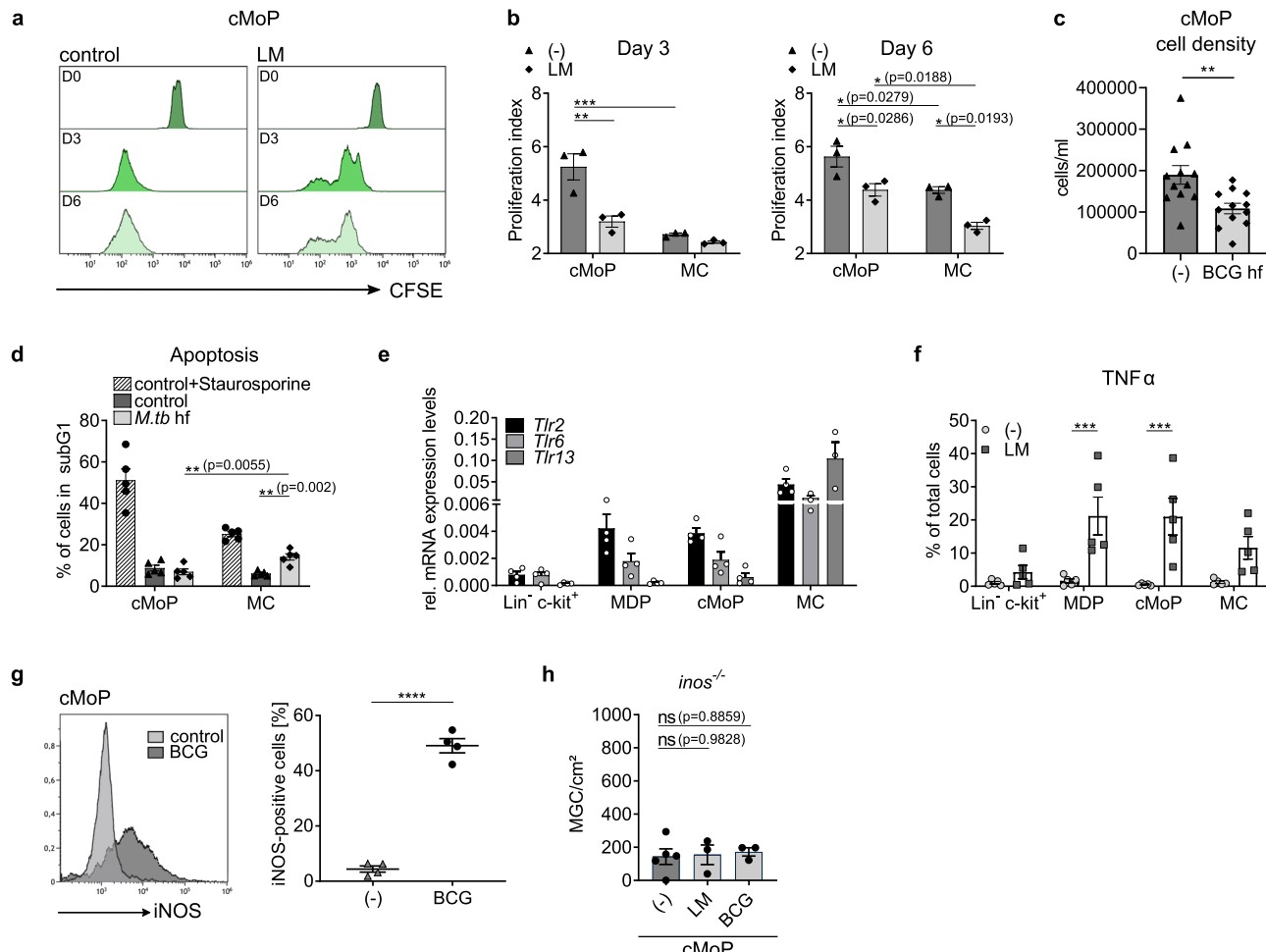

**Fig. 2 cMoP combine macrophage and progenitor characteristics. a** cMoP from WT mice were stained with CFSE and stimulated with lipomannan from *M. smegmatis* (LM, 4 µg/ml), followed by FACS analysis after 5 h (D0, dark green), 3 days (D3, green) and 6 days (D6, light green). **b** Proliferation index based on CFSE staining (Fig. 2a) was determined as $PI = \left(\sum_1^i i * \frac{N_i}{2^i}\right) / \left(\sum_1^i \frac{N_i}{2^i}\right)$ with $i$ = generation number and $N$ = number of cells in generation $i$. Bars show mean ± SEM of $n = 3$ biologically independent samples. *$p < 0.05$, **$p = 0.0028$, ***$p = 0.0007$ (two-way ANOVA, Tukey's MCT). **c** Cell density of cMoP from *Cx3cr1*$^{gfp/+}$ mice stimulated with heat-fixed (hf) BCG ($10^6$/ml) compared to M-CSF control (-) for 6 days determined with FACS counting particles. Bars show mean ± SEM of $n = 12$ biologically independent samples examined over four experiments. **$p = 0.0049$ (student's unpaired t-test, two-tailed). **d** cMoP and MC from WT mice were stimulated for 6 days with hf *M.tb* (50 µg/ml) and stained with propidium iodide to quantify the percentage of apoptotic cells in SubG1 phase. Staurosporine (1 µg/ml) added 6 h prior to analysis served as positive control. Depicted are mean ± SEM of $n = 5$ biologically independent samples. **$p < 0.01$ (two-way ANOVA, Tukey's MCT). **e** *Tlr2*, *Tlr6*, *Tlr13* mRNA expression levels in BM precursors relative to *Gapdh*. Shown are mean ± SEM of $n = 4$ (Lin$^-$ c-kit$^+$, MDP, cMoP, MC *Tlr2*) or $n = 3$ (MC *Tlr6* and *Tlr13*) biologically independent samples. **f** TNFα production of BM precursors from *Cx3cr1*$^{gfp/+}$ mice after LM stimulation (4 µg/ml) for 5 h. Depicted are mean ± SEM of n = 5 biologically independent samples. ***$p = 0.0005$ (MDP), ***$p = 0.0003$ (cMoP), (two-way ANOVA, Sidak's MCT). **g** Representative histogram and quantification of intracellular iNOS staining in cMoP from *Cx3cr1*$^{gfp/+}$ mice, stimulated with BCG (MOI 20) for 8 days. Depicted are mean ± SEM of $n = 4$ biologically independent samples examined over two experiments. ****$p < 0.0001$ (student's unpaired t-test, two-tailed). **h** Quantification of MGC formation by cMoP from *inos*$^{-/-}$ after LM stimulation (4 µg/ml) or BCG (MOI 100) for 6 days. Depicted are mean ± SEM of $n = 5$ (-) or $n = 3$ (LM, BCG) biologically independent samples. ns = not significant (one-way ANOVA, Dunnett's MCT).

between cMoP and MC with respect to their transcriptomes might also be connected with their differences in longevity, cell death, and transformation into MGC.

**Cholesterol and fatty acid biosynthesis are essential for MGC formation.** In order to dissect the role of lipid and cholesterol metabolism in the formation of MGC, we performed staining with the cholesterol and lipid dye filipin in MGC from cMoP infected with BCG-RFP. We observed the accumulation of cholesterol/lipids in MGC (Fig. 4a). Filipin staining in flow cytometry revealed that cMoP had a significantly faster accumulation of cholesterol/lipids upon stimulation with fixed BCG compared to MC, i.e., already after 2 days (Fig. 4b).

In addition, we quantified the cholesterol content in cMoP with the fluorometric amplex red assay. We found higher cholesterol concentrations in cMoP infected with BCG as compared to resting cMoP (Supplementary Fig. 3a), highlighting a net upregulation of cholesterol synthesis as a consequence of the transcriptional changes in fatty acid and cholesterol producing metabolic pathways (Figs. 3d and e). Lipid accumulation, however, requires that the cellular export systems are not upregulated in parallel to biosynthesis. The transcriptomes of resting and stimulated cMoP did not unveil downregulation of

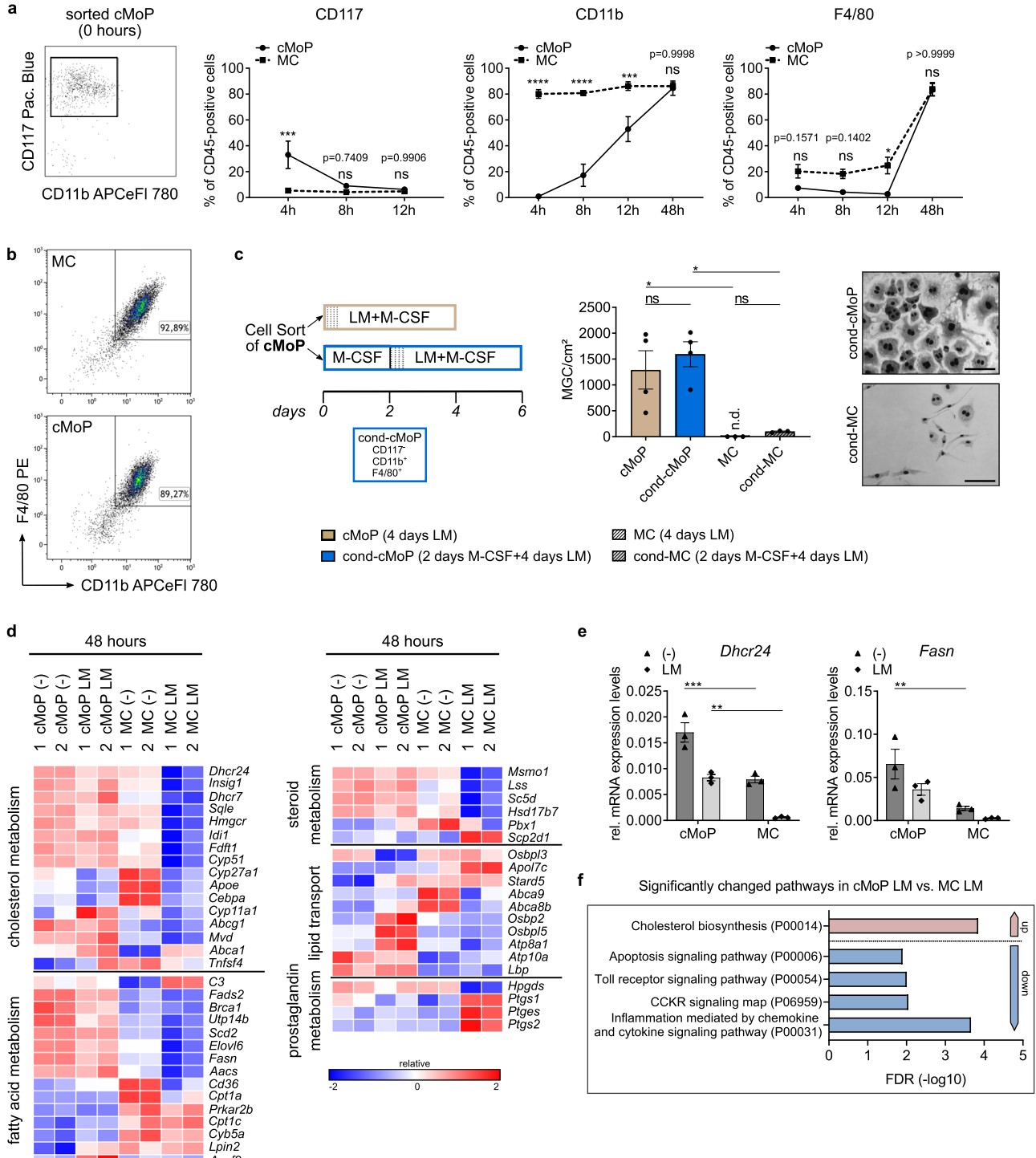

*Abcg1* and *Abca1*, genes coding for the two main cholesterol transporters (Fig. 3d). However, this analysis was performed 48 h after induction of the MGC transformation, i.e., before MGC have fully developed (compare Fig. 1d). In order to complement this analysis, mRNA expression levels of *Abca1* and *Abcg1* were established 6 days following differentiation. At this time point these genes were downregulated in cMoP-derived MGC (Fig. 4c). Accordingly, it seemed safe to conclude that the accumulation of cholesterol/lipids originated from dysregulated metabolic path-ways and, additionally, a reduction in cholesterol and triglycer-ides efflux. We tested whether cholesterol accumulation was, besides being a signature trait of cMoP, causally linked to the

MGC transformation process. Accordingly, we interfered with the cholesterol metabolism in MGC during the transformation process and analyzed the effect on the phenotype. First, we employed cholesterol oxidase, a flavoenzyme that associates with lipid bilayers, binds cholesterol and desorbs it from the membrane. Coincubation of cMoP with cholesterol oxidase during the transformation process decreased MGC formation by about 50% when stimulated with live or fixed bacteria (Fig. 4d and Supplementary Fig. 3b). In addition, cholesterol was extracted from the cells with methyl-β-cyclodextrin (MBCD), which facilitates the efflux of cholesterol by complexing and solubilizing the lipids[26]. In line with the effects of cholesterol

**Fig. 3 cMoP-derived MΦ keep their potential to form MGC and show a characteristic metabolic profile. a** Dot plot of cMoP analyzed immediately after isolation for initial surface marker expression. Surface marker expression of cMoP and MC from *Cx3cr1$^{gfp/+}$* mice analyzed by FACS related to CD45$^+$ cells after 4, 8, 12 or 48 hours in culture with M-CSF. Depicted are mean ± SEM of *n* = 3–4 (CD117), *n* = 5–9 (CD11b), *n* = 3–5 (F4/80) biologically independent samples. Detailed sample numbers (*n*) are presented in Supplementary Table 4. ns = not significant, *p = 0.0161, ***p = 0.0009 (CD117), ***p = 0.0005 (CD11b), ****p < 0.0001 (two-way ANOVA, Sidak's MCT). **b** Representative FACS dot plots for F4/80 and CD11b expression of cMoP and MC after 48 h in medium supplemented with M-CSF. **c** cMoP or MC from *Cx3cr1$^{gfp/+}$* mice were stimulated either directly with LM (4 μg/ml) for 4 days or after 2 days in culture with M-CSF (cond-cMoP, cond-MC). MGC formation was quantified based on Hemacolor stainings (scale bar: 100 μm). Bars show mean ± SEM of *n* = 4 (cMoP) and *n* = 3 (MC) independent biological samples. ns = not significant (cMoP — cond-cMoP: *p* = 0.8162; MC — cond-MC: *p* = 0.9942), *p = 0.0256 (cMoP — MC), *p = 0.0109 (cond-cMoP — cond-MC), (one-way ANOVA, Tukey's MCT). **d** Heatmaps of gene expression in cMoP and MC from *Cx3cr1$^{gfp/+}$* mice after LM stimulation (4 μg/ml, 48 hours) with M-CSF (50 ng/ml). M-CSF alone served as control (-). Depicted are standardized log2 differences of significantly up- (red) or downregulated (blue) genes in cMoP-LM compared to MC-LM or cMoP (-) compared to MC (-). The analysis was performed with two biologically independent samples per group (1 = sample 1; 2 = sample 2) by gene-ontology terms (GO-Terms), (GO:0006631 fatty acid metabolic process, GO:0008203 cholesterol metabolic process, GO:0001516 prostaglandin biosynthetic process, GO:0006694 steroid biosynthetic process, GO:0006869 lipid transport). If GO-Terms overlapped, respective genes were only assigned to one group. **e** *Dhcr24* and *Fasn* mRNA expression levels relative to *Gapdh* in cMoP and MC, stimulated with LM (12 μg/ml, light gray, diamonds) for 48 h compared to M-CSF control (dark gray, triangles) determined by qRT-PCR. Bars show mean ± SEM of *n* = 3 biologically independent samples. **p = 0.0015 (*Dhcr24*), **p = 0.0091 (*Fasn*), ***p = 0.0005 (two-way ANOVA, Sidak's MCT). **f** Panther pathway analysis of gene expression data described in Fig. 3d. Depicted are significantly up- (red) or downregulated (blue) pathways in cMoP-LM versus MC-LM.

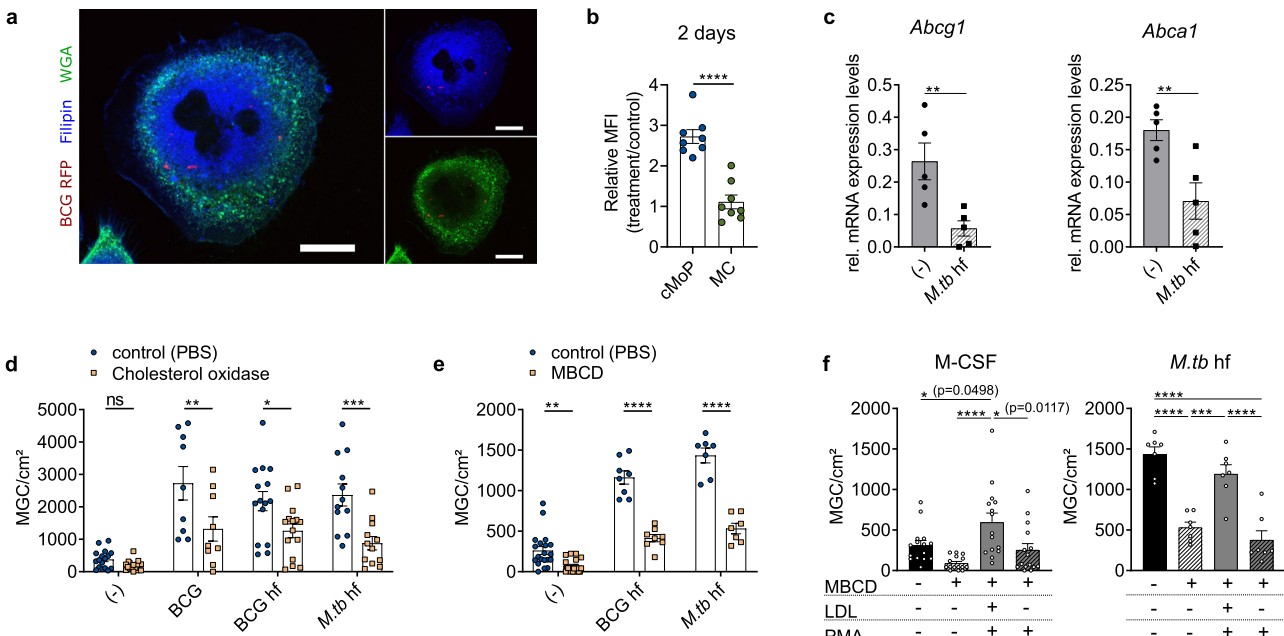

**Fig. 4 Cholesterol metabolism is an essential prerequisite for MGC formation. a** Filipin (blue) and wheat germ agglutinin (WGA, green) staining of cMoP-derived MGC induced with heat-fixed (hf) BCG (10$^6$/ml) and additionally infected with BCG-RFP (MOI 1, red). Scale bar: 20 μm. Representative staining out of three independent experiments shown. Channels were adjusted individually with respect to contrast and brightness. **b** Filipin staining, depicted is the mean fluorescence intensity (MFI) of cMoP (blue) and MC (green) from *Cx3cr1$^{gfp/+}$* mice stimulated with hf BCG (10$^6$/ml) (treatment) normalized to M-CSF control at day 2. Shown are mean ± SEM of *n* = 8 biologically independent samples examined over three experiments. ****p < 0.0001 (student's unpaired *t*-test, two-tailed). **c** *Abcg1* and *Abca1* mRNA expression levels relative to *Gapdh* mRNA of cMoP stimulated with heat-fixed *M.tb* (50 μg/ml) for 6 days determined by qRT-PCR. Bars show mean ± SEM of *n* = 5 biologically independent samples. **p = 0.0099 (*Abcg1*), **p = 0.0095 (*Abca1*), (student's unpaired *t*-test, two-tailed). **d** cMoP from *Cx3cr1$^{gfp/+}$* mice were stimulated with BCG (MOI 20), heat-fixed BCG (10$^6$/ml) or heat-fixed *M.tb* (10 μg/ml) and treated with cholesterol oxidase (2 mU/ml, beige) or PBS (vehicle, blue) for 6 days. MGC quantification by Hemacolor staining. Bars show mean ± SEM of *n* = 17 (-), *n* = 9 (BCG), *n* = 15 (BCG hf), *n* = 12 (*M.tb*) independent biological samples examined over 11 experiments. ns = not significant (*p* = 0.9504), *p = 0.0251, **p = 0.0052, ***p = 0.0004 (two-way ANOVA, Sidak's MCT). **e** cMoP from *Cx3cr1$^{gfp/+}$* mice stimulated with heat-fixed BCG (10$^6$/ml) or heat-fixed *M.tb* (10 μg/ml) and treated with methyl-β-cyclodextrin (MBCD, 1 mmol, beige) or dH$_2$O (vehicle, blue) for 6 days. MGC quantification by Hemacolor staining. Bars show mean ± SEM of *n* = 20 (-), *n* = 8 (BCG), *n* = 7 (*M.tb*) biologically independent samples. **p = 0.0038, ****p < 0.0001 (two-way ANOVA, Sidak's MCT). **f** cMoP from *Cx3cr1$^{gfp/+}$* mice were stimulated with heat-fixed *M.tb* (10 μg/ml) and treated with methyl-β-cyclodextrin (MBCD, 1 mmol), LDL (50 μg/ml) or PMA (1 μg/ml) for 6 days. Vehicle control for MBCD was dH$_2$O. Bars show mean ± SEM of *n* = 15 biologically independent samples in seven experiments (M-CSF control) and *n* = 7 biologically independent samples in three experiments (*M.tb*). *p < 0.05, ***p = 0.0004, ****p < 0.0001 (one-way ANOVA, Tukey's MCT).

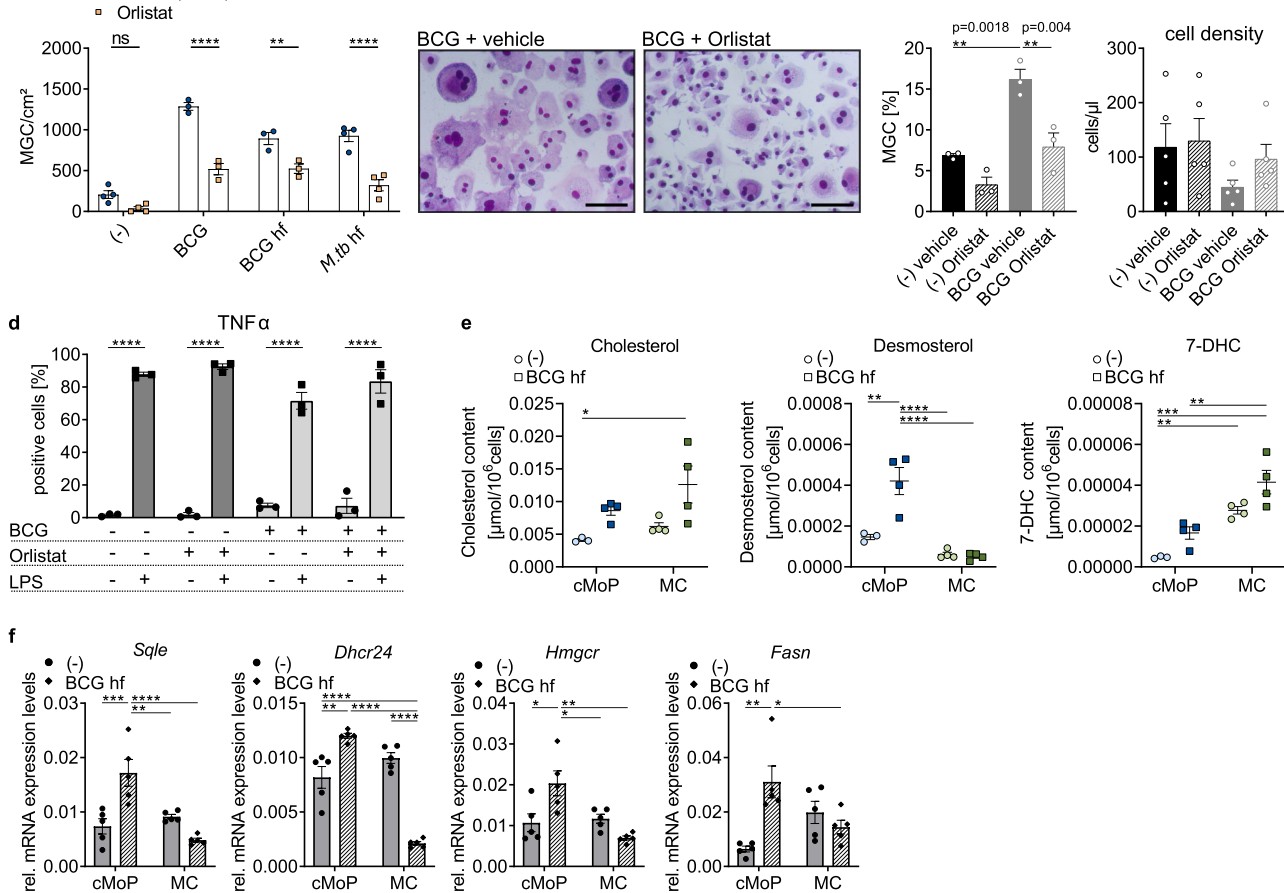

**Fig. 5 Altering fatty acid metabolism by orlistat affects MGC formation. a** cMoP from $Cx3cr1^{gfp/+}$ mice were stimulated with BCG (MOI 20), heat-fixed BCG ($10^6$/ml), heat-fixed $M.tb$ (50 μg/ml) and treated with orlistat (100 μmol, beige) or vehicle control (ethanol, blue) for 6 days. MGC were quantified by Hemacolor staining (scale bar: 100 μm). Gamma-correction was adjusted to 0.45 in both images. Bars depict mean ± SEM of $n = 3$ (BCG, BCG hf) and $n = 4$ (control, $M.tb$) biologically independent samples. ns = not significant ($p = 0.1532$), **$p = 0.0022$, ****$p < 0.0001$ (two-way ANOVA, Sidak's MCT). **b** Hoechst staining of cMoP from $Cx3cr1^{gfp/+}$ mice stimulated with BCG (MOI 20) and treated with orlistat (100 μmol) or ethanol (control) for 6 days. MGC were quantified as polyploid cells ($N > 4$) from histogram peaks. Bars show mean ± SEM of $n = 3$ biologically independent samples. **$p < 0.01$ (one-way ANOVA, Tukey's MCT). **c** Cell density determined by FACS counting particles of cMoP from $Cx3cr1^{gfp/+}$ mice treated as indicated in Fig. 5b. Bars show mean ± SEM of $n = 5$ biologically independent samples. **d** TNFα production of cMoP from $Cx3cr1^{gfp/+}$ mice treated as indicated in Fig. 5b. LPS (100 ng/ml) was added 5 h prior to analysis. Bars show mean ± SEM of $n = 3$ biologically independent samples. ****$p < 0.0001$ (one-way ANOVA, Sidak's MCT). **e** Sterol analysis of cMoP (blue) and MC (green) from $Cx3cr1^{gfp/+}$ mice stimulated with heat-fixed BCG ($10^6$/ml, squares) for 6 days compared to M-CSF (50 ng/ml, circles) control (-). Shown are cholesterol, desmosterol, and 7-dehydrocholesterol (7-DHC) content normalized to cell number. Depicted are mean ± SEM of n = 4 biologically independent samples (cMoP control with $n = 3$ as one sample could not be analyzed for technical reasons). *$p = 0.0195$, **$p = 0.0019$ (desmosterol), **$p = 0.002$ (7-DHC, BCG), **$p = 0.0063$ (7-DHC, control), ***$p = 0.0001$, ****$p < 0.0001$ (two-way ANOVA, Tukey's MCT). **f** $Sqle$, $Dhcr24$, $Hmgcr$ and $Fasn$ mRNA expression levels of cMoP and MC stimulated with heat-fixed BCG ($10^6$/ml, diamonds, hatched bars) for 6 days relative to $Gapdh$ compared to M-CSF (50 ng/ml) controls (circles, gray bars). Bars show mean ± SEM of $n = 5$ biologically independent samples in two experiments. *$p = 0.0328$ ($Fasn$), *$p = 0.0155$ ($Hmgcr$, cMoP-control — cMoP-BCG), *$p = 0.0313$ ($Hmgcr$, MC-control — cMoP-BCG), **$p = 0.0057$ ($Sqle$), **$p = 0.0013$ ($Dhcr24$), **$p = 0.001$ ($Hmgcr$), **$p = 0.0016$ ($Fasn$), ***$p = 0.001$, ****$p < 0.0001$ (two-way ANOVA, Tukey's MCT).

oxidase, MBCD substantially inhibited MGC formation (Fig. 4e and Supplementary Fig. 3c). We then employed the reverse approach by adding human low-density lipoproteins (LDL) to the culture. In order to facilitate the uptake of LDL, PMA was added[27]. Using this approach, we found LDL donation to rescue MGC formation in MBCD treated cells (Fig. 4f and Supplementary Fig. 3d). As a complementary approach, we used tetrahydrolipstatin (orlistat), which irreversibly inhibits the fatty acid synthase[28] and prevents triacylglycerol degradation[29]. In full support of the model that lipid provision is rate limiting for the MGC transformation process by cMoP, we found orlistat to inhibit MGC formation in response to fixed and live mycobacteria (Fig. 5a). Orlistat reduced the total number of nuclei per MGC and the percentage of polyploid cells ($N > 4$) (Fig. 5b).

Moreover, orlistat prevented the foamy, lipid-rich cytoplasmic appearance and conserved relatively small MΦ with several pseudopodia (Fig. 5a). Since interfering with lipid metabolism is potentially impacting on cell viability, we carefully assessed any toxic effects of cholesterol oxidase, MBCD and orlistat. We found that cMoP incubated with BCG and orlistat proliferated more than cMoP treated with BCG alone and were similar to untreated control cells in this respect (Fig. 5c). In full agreement, orlistat treated cells did not differ from untreated cells with respect to TNFα formation in response to LPS (Fig. 5d). Upon addition of cholesterol oxidase, cell density and TNFα production after LPS stimulation were not affected compared to vehicle controls (Supplementary Figs. 3e and f). When measuring cell density for MBCD stimulated cells, we found a reduction, when LDL was

added additionally, which was in line with the high amount of MGC formed under these conditions (Supplementary Fig. 3g). The reduction in cell number upon addition of MBCD alone to *M.tb* stimulated cMoP (Supplementary Fig. 3g) was likely due to the reduced availability of cholesterol for membrane formation and proliferation. The efficacy of MBCD to extract cholesterol from the cells was examined by filipin staining (Supplementary Fig. 3h). Similar to cholesterol oxidase and orlistat, MBCD did not impact on the TNFα response to LPS (Supplementary Fig. 3i). Together, these findings suggested that blocking of the fatty acid synthase and depriving the cells of cholesterol inhibit the MGC transformation process driven by mycobacteria, and conserves the mononuclear MΦ state. Next, we dissected the differences in lipid composition of BCG-stimulated cMoP and MC by liquid chromatography-mass spectrometry-based sterol analyses. After 6 days of BCG stimulation, we found cholesterol accumulation in both cMoP and MC. However, the accumulation of cholesterol precursors significantly differed in these cell types. BCG-stimulated cMoP showed an increase in desmosterol content, in contrast to MC, which rather contained 7-dehydrocholesterol (7-DHC) (Fig. 5e and Supplementary Fig. 3j). Both sterols are precursors of cholesterol. Desmosterol is formed via the Bloch pathway, and 7-DHC via the Kandutsch-Russell pathway[30]. Previously, the accumulation of desmosterol was reported to be characteristic for foam cell transformation in atherosclerosis-associated MΦ[31]. Furthermore, an immunomodulatory function was described for specific cholesterol precursor derivatives[32]. Accordingly, the accumulation of different cholesterol precursors in BCG-stimulated cMoP and MC indicated that cholesterol metabolism was differentially regulated in these subsets. In order to explore, whether cMoP keep cholesterol biosynthesis upregulated during the entire transformation process into MGC, we analyzed the expression of key enzymes of the cholesterol and fatty acid biosynthesis. We found upregulation of *Sqle*, *Dhcr24*, *Hmgcr*, *Fasn*, *Acat2,* and *Cyp51* in BCG-stimulated cMoP in contrast to MC (Fig. 5f and Supplementary Fig. 3k). However, the expression of *Acat1* involved in β-oxidation and the degradation of fatty acids was downregulated in stimulated cells (Supplementary Fig. 3k). In addition, the upregulation of cholesterol biosynthesis in cMoP-derived MGC in contrast to monocytes was in line with the transcriptome analysis for LM-stimulated cMoP (Fig. 3f).

In order to further dissect the intrinsic metabolic signature in MGC, we analyzed the glycolytic phenotype in cMoP-derived MGC using an extracellular flux analyzer. MGC showed a higher extracellular acidification rate in glycolysis stress tests than control cells (Supplementary Fig. 4a), as well as a significant upregulation of glycolysis. This indicated that in addition to fatty acid and cholesterol metabolism, glycolysis plays an important role in MGC transformation, which hints at high ATP consumption requirements. These observations are in full accordance with a decrease in MGC formation in the presence of 2-deoxy-D-glucose (2-DG), a competitive inhibitor of glycolysis (Supplementary Fig. 4b). Since the target of rapamycin (mTOR) has previously been implicated in the switch from oxidative phosphorylation to glycolysis in DC[33], we tested the impact of rapamycin on MGC formation and found a decrease of MGC transformation in cMoP (Supplementary Figs. 4c and d). Together, these findings emphasize the close interaction between the immune response to mycobacteria/TLR2 ligands and metabolic changes in cMoP.

### iMoP — a circulating monocyte progenitor in *M.tb* infection.
After establishing the high potential of cMoP to form MGC, we set out to define their role in vivo. We employed two different

mouse models for pulmonary tuberculosis. Aerogenic exposure to a low-dose inoculum of virulent *M.tb* results in progressive disease, with a peak of infection after 21 days, in both the resistant mouse strain C57BL/6 and in the susceptible strain 129S2. Whereas C57BL/6 mice survive and develop a chronic infection, 129S2 mice succumb to infection within 5 weeks and mimic active tuberculosis in humans[34,35]. In both mouse strains, we found increased frequencies of cMoP in the bone marrow at day 21 post-infection. In contrast, cMoP were hardly detectable in the blood during tuberculosis progression (Fig. 6a). We postulated that cMoP undergo early changes with respect to their cell surface marker expression, similar to our in vitro observations (Fig. 3a). In line with this assumption, we identified a circulating cell population that differed from cMoP only with respect to CD117 expression (Fig. 6b). This cell type, which we denominated induced monocyte progenitors (iMoP), fully correlated with the differentiation phenotype of cMoP in vitro, where 70% of cells lost CD117 expression within 4 hours. Similar to cMoP, we found iMoP to be present in the bone marrow of uninfected mice, and an increase of iMoP frequency at day 21 in infected mice. In contrast to cMoP, which almost exclusively reside in bone marrow and spleen, iMoP were detectable at a low frequency in peripheral blood already in steady state. During infection, iMoP markedly increased up to 1% of CD45-positive blood cells at day 14 and 21, especially in the susceptible 129S2 mice (Fig. 6c). Hemacolor staining of freshly sorted progenitors showed iMoP with decreased nucleus to cytoplasma ratio compared to MDP and cMoP, but higher cellular granularity (Fig. 6d). We further assessed whether iMoP retained the capacity to form MGC. We extracted cMoP, iMoP and MC from the bone marrow of *Cx3cr1*[gfp/+] mice. iMoP showed an intermediate potential to produce MGC upon exposure to both live and fixed bacteria (Fig. 6e). In contrast to MDP, which also showed an intermediate potential to form MGC, iMoP more specifically formed multinucleated giant cells (Supplementary Fig. 5a). Furthermore iMoP, cMoP, and MC produced TNFα within 5 h after stimulation with fixed *M. bovis* BCG, whereas more immature progenitors like Lin⁻ c-kit⁺ cells and MDP did not show this ability (Fig. 6f). To further analyze the recruitment of iMoP into the blood stream, we infected 129S2 mice with a low dose of *M.tb* and neutralized CCR2-positive cells with the application of a CCR2 antibody (MC-21) starting day 14 p.i.. Mice injected with the CCR2-specific antibody showed a significant decrease of iMoP frequency in the BM and completely failed to mobilize iMoP into the blood (Fig. 6g and Supplementary Fig. 5b). We also observed a significant reduction of cMoP frequency in the BM after antibody treatment (Fig. 6g). These results were in line with our flow cytometry analysis of CCR2 expression on the different bone marrow progenitor subsets, where we observed increased binding of the MC-21 antibody depending on the maturity state of the progenitor subsets (Supplementary Fig. 5c). Injection of the CCR2-specific antibody did not influence bacterial burdens in the lung or spleen (Supplementary Fig. 5d). Accordingly, circulating iMoP were not essential to control *M.tb* in these mouse models, where MGC can hardly be found at the investigated time points (Supplementary Figs. 5e, f and g). However, the spontaneous formation of neutralizing antibodies against MC-21 antibodies in treated mice precludes longer depletion experiments. Thus, in order to further explore the role of CCR2 in MGC formation at later stages of infection, we analyzed *M.tb* infected *Ccr2*⁻/⁻ mice from day 60 to day 300 p.i.. We found an increase in the total area of inflammation in lungs from *Ccr2*⁻/⁻ mice with time, which was not visible in wild-type infected mice (Supplementary Figs. 5e, f and g). This is compatible with the notion that circulating monocyte progenitors and/or mature monocytes contribute

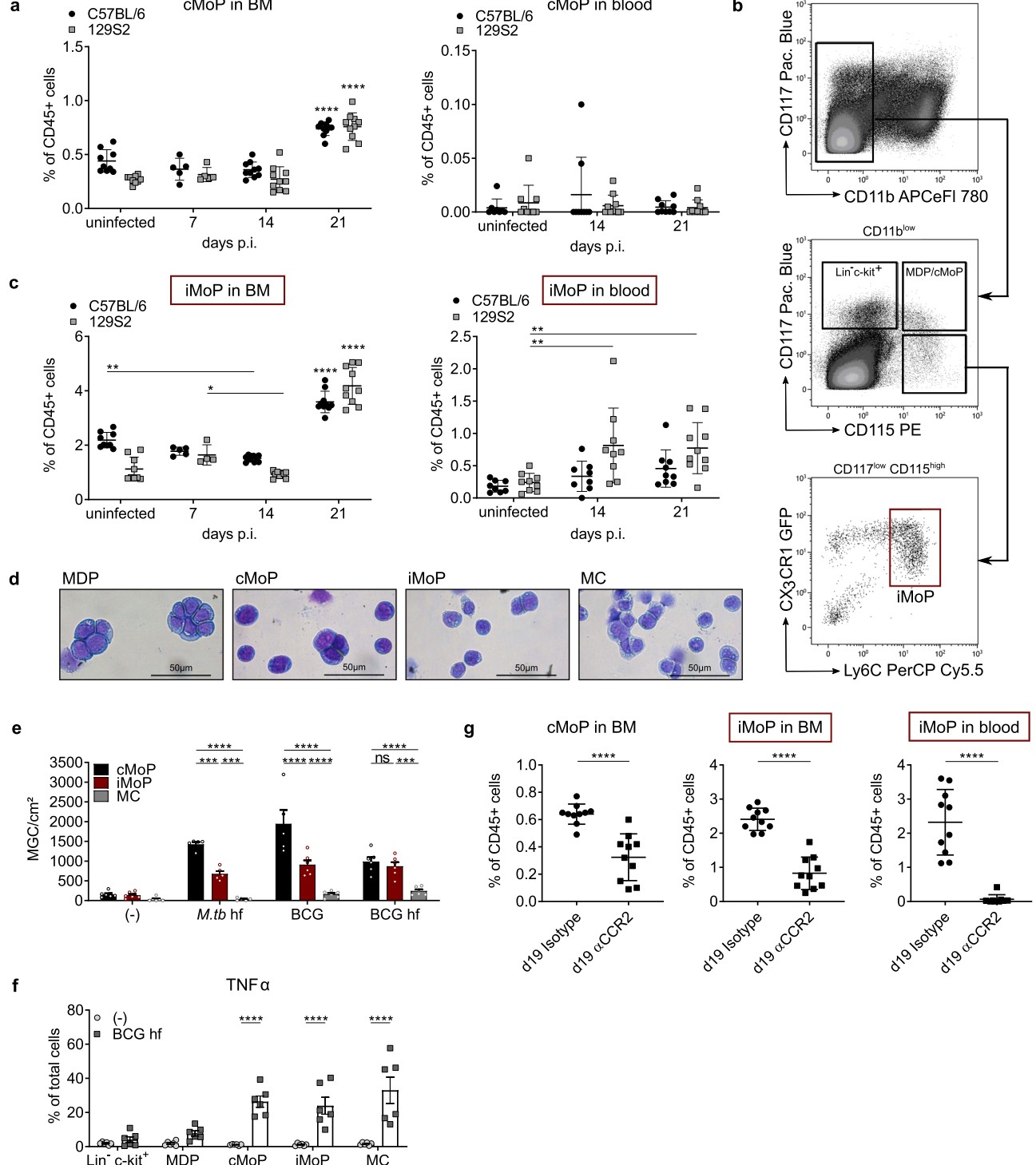

**d**  MDP          cMoP          iMoP          MC

*(microscopy panels with 50μm scale bars)*

to inflammation control in pulmonary Tb infection. In contrast, the relative immune cell composition was largely similar in both mouse strains, including MGC with high lipid content and foam cell formation after 100 and 300 days of infection (Supplementary Fig. 5f). Of note, monocytes (and probably their precursors) are reduced by 80–90% in $Ccr2^{-/-}$ mice[36]. Thus, the remaining monocyte progenitors in the circulation of $Ccr2^{-/-}$ mice might fill the MGC progenitor niche during the mycobacterial infection, in particular since chronic lung inflammation in *M.tb* infection is a slow process with relatively low cell turnover. Notably, isolated progenitors from wild-type and $Ccr2^{-/-}$ mice treated with heat-fixed BCG exhibited a largely similar potential to form MGC

in vitro, with only a modestly reduced capacity of CCR2-deficient cMoP (Supplementary Figs. 5h and i).

In summary, we identified a circulating monocyte progenitor subset termed iMoP, which has a high capacity to form MGC and circulates at increased frequency in mycobacterial infection.

**iMoP — a progenitor cell between cMoP and monocytes.** The surface marker expression ($CX3CR1^+$ $CD117^-$ $CD115^+$ $Ly6C^+$ $CD11b^-$) of iMoP differed from cMoP with respect to CD117 expression and from MC with respect to the lack of CD11b expression. CD115 upregulation classified iMoP as part of the monocyte/macrophage lineage. Since, in contrast to cMoP, iMoP

**Fig. 6 iMoP – a circulating monocyte precursor in *M.tb* infection. a**, **c** Frequency of cMoP and iMoP in bone marrow (BM) and blood of mice infected via aerosol with *M.tb* (H37Rv, 200 CFU) analyzed by FACS 7, 14 and 21 days post infection (p.i.). Graphs show mean ± SD of $n = 8$–9 mice (uninfected controls), $n = 8$–10 mice (day 14, day 21) in two experiments and $n = 4$–5 mice (day 7) in one experiment. Detailed numbers of mice ($n$) for each condition/time point are presented in Supplementary Table 5. *$p = 0.0125$, **$p = 0.0017$ (iMoP in BM), **$p = 0.0022$ (iMoP in blood, D14), **$p = 0.0036$ (iMoP in blood, D21), ****$p < 0.0001$ (refers to all other time points of the respective mouse line), (two-way ANOVA, Tukey's MCT). Two outliers were removed from iMoP in blood at day 14 C57BL/6 (2.49) and 129S2 (8.50), calculated by Grubbs test. **b** Flow cytometry gating of iMoP from BM of *Cx3cr1*$^{gfp/+}$ mice. **d** Hemacolor staining of murine BM precursors immediately after cell sorting. Scale bar: 50 μm. Gamma-correction was adjusted (0.53) in all images. **e** MGC formation of cMoP, iMoP (red) and MC stimulated with BCG (MOI 20), heat-fixed BCG ($10^6$/ml) or heat-fixed *M.tb* (10 μg/ml) for 6 days. Bars show mean ± SEM of $n = 5$ (*M.tb*, cMoP-BCG) or $n = 6$ (control, BCG hf, iMoP-BCG, MC-BCG) biologically independent samples examined over three experiments. ns = not significant ($p = 0.7241$), ***$p = 0.0001$ (*M.tb*: cMoP – iMoP), ***$p = 0.0009$ (*M.tb*: MC – iMoP), ***$p = 0.0004$ (BCG hf: MC – iMoP), ****$p < 0.0001$ (two-way ANOVA, Tukey's MCT). **f** TNFα production of BM precursors from *Cx3cr1*$^{gfp/+}$ mice after stimulation with heat-fixed BCG ($10^6$/ml) for 5 h determined by FACS compared to M-CSF controls. Depicted are mean ± SEM of $n = 6$ independent biological samples examined over two experiments. ****$p < 0.0001$ (two-way ANOVA, Sidak's MCT). **g** *M.tb* aerosol infection (H37Rv, 200 CFU) of 129S2 mice, followed by intraperitonal injection of a CCR2-specific antibody (MC-21, 20 μg daily) or isotype control (MC-67), starting on day 14 p.i.. FACS analysis of progenitor frequency in blood and BM at day 19 p.i.. Graphs show mean ± SD of $n = 10$ mice per group examined over two experiments. ****$p < 0.0001$ (student's unpaired *t*-test, two-tailed).

were capable of circulating in the blood, we aimed to further characterize this so far undescribed bone marrow subset and to distinguish these cells from other progenitors. We therefore performed a transcriptomic analysis of iMoP and cMoP, sorted directly from murine bone marrow and found 391 genes to be significantly upregulated, and 260 genes to be significantly downregulated in iMoP as compared to cMoP (Fig. 7a). The analysis revealed that cell cycle and cell differentiation were differentially regulated in iMoP compared to cMoP, which corresponded to their different progenitor status. *Kit oncogene* (CD117) was clearly downregulated in iMoP, in line with our previous flow cytometry analysis. In contrast to this, the gene expression of hematopoetic cell kinase (*Hck*), important for regulation of immune response in myeloid cells and reported to be involved in cell recruitment to the lung and cytokine production[37], was increased in iMoP (Fig. 7b). Other genes, e.g., *Tlr13*, *C3*, *Cd180*, and *Il1rn*, which are involved in innate immune response and cytokine-mediated signaling, were upregulated in iMoP. The upregulation of *Trem2*, which is expressed in iMoP, is reported to be connected with immune response to mycobacterial infections[38]. This spurs the hypothesis that iMoP are involved in mycobacterial immunity. In addition, we found differences between iMoP and cMoP with respect to cell adhesion and cell migration, which might explain their various potential to circulate in the blood (Fig. 7b). Gene ontology enrichment analysis of genes overexpressed in iMoP compared to cMoP yielded terms such as positive regulation of monocyte chemotaxis, regulation of inflammatory response and regulation of phagocytosis, which further supported iMoP to be part of the monocyte/macrophage lineage (Fig. 7c). Although the transcriptomic analysis indicated an increased expression of *Itgam* (CD11b) in iMoP as compared to cMoP (Fig. 7b), quantitative analysis (qRT-PCR) analysis of sorted lineages revealed that compared to MC, iMoP were low for *Itgam*, as well as *Adgre1* (F4/80). For *Mpo*, *Tlr13* and *Msr1*, iMoP showed intermediate expression levels compared to cMoP and MC (Fig. 7d). These results are in line with our hypothesis that iMoP represent a circulating monocyte progenitor subset, which originates in the bone marrow. In contrast to cMoP, iMoP did not form colonies in MethoCult assay. Instead, they rather formed groups of 2–3 cells, indicating that they lost their stem cell potential and only undergo a limited number of divisions (Fig. 7e). We further analyzed the capacity of the different subsets to kill intracellular bacteria and found more live BCG in MC as compared to cMoP and iMoP 4 days post infection (Supplementary Fig. 5j), although all cell types failed to eliminate mycobacteria. It has been reported before that cMoP show a high expression of *Ms4a3*[13]. Thus we examined *Ms4a3* expression

levels in the progenitor subsets and found cMoP to have the highest expression. MC from the bone marrow, defined by their CD11b expression, did not express *Ms4a3*, whereas iMoP and MDP showed distinctly lower levels compared to cMoP (Fig. 7f). To confirm that iMoP can give rise to CD11b positive monocytes in vivo, we adoptively transferred iMoP, which had been isolated from bone marrow of *β-actin gfp*$^{+/-}$ mice, i.v. (Fig. 7g). After 14 h all transferred iMoP still expressed Ly6C. In the blood, almost all GFP$^+$ cells had upregulated CD11b. In the spleen and bone marrow, around 70% showed CD11b expression, whereas F4/80 expression was still rather low in all compartments (Figs. 7g and h). These observations confirmed that iMoP isolated from the bone marrow could indeed give rise to inflammatory monocytes. Together these data indicate, that iMoP are a bone marrow precursor subset between cMoP and CD11b-positive monocytes, which can give rise to circulating monocytes.

**Transferred bone marrow precursors migrate to spleen and liver granulomas in *M. bovis* BCG infected mice.** Next we addressed whether immature bone marrow cells localize to the site of mycobacterial infections and become part of the granuloma. Therefore, we made use of an established model, where *M. bovis* BCG i.v. infection leads to granuloma formation in liver and spleen (Fig. 8a). BCG infection resulted in substantial splenomegaly at day 18 and to a lesser extent at day 30 post infection (Figs. 8a and b), with CD68-positive, BCG containing granulomas in the spleen (Fig. 8c) and liver. In accordance with our data on pulmonary *M.tb* infection, CD11b$^-$, Ly6C$^+$, CD115$^+$ cells (including cMoP and iMoP and excluding MC) were increased in spleen and bone marrow. With respect to the kinetics, these cells were most frequent at day 18, at the peak of infection, but they stayed high in number until day 30, especially in the bone marrow (Fig. 8d and Supplementary Fig. 6a). In spleen and blood, Ly6C$^+$ monocytes were transiently increased at day 18 post infection, but they decreased to steady state levels until day 30. Interestingly, the monocyte compartment in the bone marrow was not significantly affected in BCG infection, in contrast to the precursor subsets (Supplementary Fig. 6b). In addition, granulocytes increased during the infection, whereas B-cells slightly decreased at day 18 post infection referred to all CD45-positive cells. The proportions of the T-cell subsets were not altered during the time course of infection (Supplementary Fig. 6c). Next, we made use of the BCG infection model to further study the role of precursor cells in *Ccr2*$^{-/-}$ mice during infection. As outlined above, the recruitment of monocyte progenitors is CCR2-dependent (Fig. 6g). In contrast to the application of the CCR2 antibody MC-21, which depletes all CCR2-expressing cells

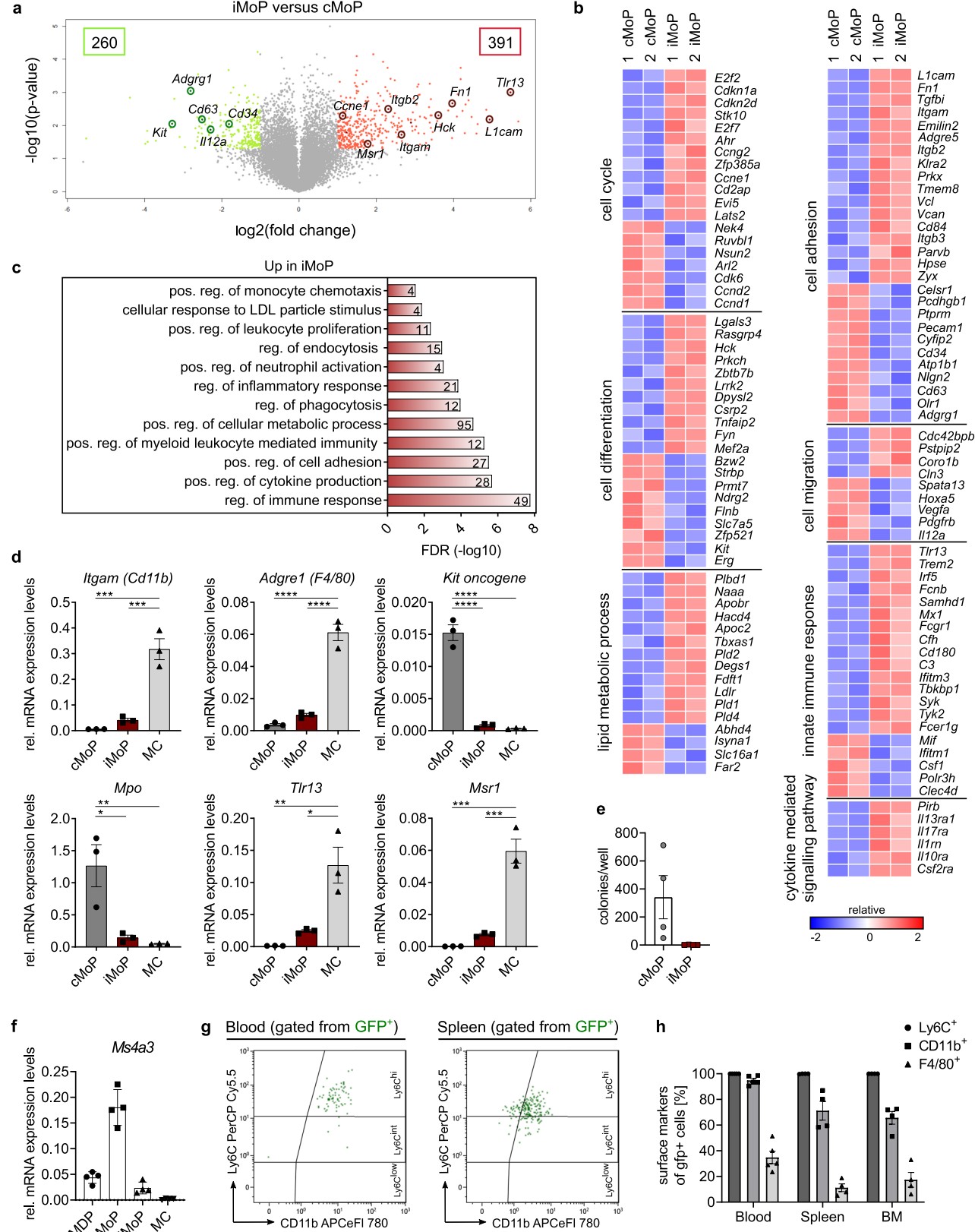

including those in the bone marrow, we found an accumulation of bone marrow progenitors both in wild-type and CCR2-deficient mice at day 18 post infection (Fig. 8e). In line with the fact that recruitment of monocytes and their progenitors is reduced in $Ccr2^{-/-}$ mice, we observed only a minimal increase in CD115+ CD11b− Ly6C+ cells in the spleens of $Ccr2^{-/-}$ as compared to wild-type mice. In BCG infected $Ccr2^{-/-}$ mice, Ly6C$^{high}$ monocytes were reduced by over 80% in the blood stream and the spleen as compared to wild-type mice. In contrast, bone marrow Ly6C$^{high}$ monocytes were present at similar frequencies in $Ccr2^{-/-}$ and control mice (Fig. 8f and Supplementary Fig. 6b). In order to explore, whether monocyte progenitors, in

**Fig. 7 iMoP – a precursor cell between cMoP and monocytes. a** Volcano plot of Clariom S gene array analysis of cMoP and iMoP. Depicted are −log10 (p-value) on the y-axis versus log2(fold change) on the x-axis. Shown are significantly upregulated (red) and downregulated (green) genes in iMoP compared to cMoP. **b** Heatmaps of gene expression in cMoP and iMoP from $Cx3cr1^{gfp/+}$ mice. Depicted are standardized log2 differences of significantly up- (red) or downregulated (blue) genes. The analysis was performed by GO-Terms (GO:0045087 innate immune response, GO:0030154 cell differentiation, GO:0007155 cell adhesion, GO:0006629 lipid metabolic process, GO:0016477 cell migration, GO:0019221 cytokine-mediated signaling pathway, GO:0007049 cell cycle). If GO-Terms overlapped, respective genes were only assigned to one group. Affymetrix Clariom S gene array was performed with two biologically independent samples per group (1 = sample 1; 2 = sample 2). **c** Compilation of over-represented gene-ontology-processes with respect to the significantly upregulated genes in iMoP compared to cMoP. **d** qRT-PCR analysis of *Itgam* (Cd11b), *Adgre1* (F4/80), *Mpo*, *Tlr13*, *Msr1* and *Kit oncogene* mRNA relative to *Gapdh* mRNA in cMoP, iMoP (red) and MC. Bars show mean ± SEM with n = 3 biologically independent samples. *p = 0.0101 (*Tlr13*), *p = 0.0142 (*Mpo*), **p = 0.0038 (*Tlr13*), **p = 0.0098 (*Mpo*), ***p = 0.0002 (*Itgam, Msr1* cMoP – MC), ***p = 0.0004 (*Itgam, Msr1* iMoP – MC), ****p < 0.0001 (ordinary one-way ANOVA, Tukey's MCT). **e** cMoP and iMoP (red) were cultured in M3434 Methocult medium. Colonies consisting of at least 10 cells were counted at day 10. Depicted are means ± SEM of n = 4 biologically independent samples examined over two experiments. **f** qRT-PCR analysis of *Ms4a3* mRNA relative to *Gapdh* in MDP, cMoP, iMoP and MC. Bars show mean ± SEM of n = 4 biologically independent samples examined over at least two experiments. **g** iMoP isolated from $\beta$-actin-gfp$^{+/-}$ mice were transferred intravenously into WT mice. Blood, spleen and bone marrow were analyzed for GFP$^+$ cells (green) 14 h post transfer by FACS for Ly6C, CD11b and F4/80 expression. Representative FACS plots are shown. **h** Quantification of FACS analysis described in Fig. 7g. Bars show mean ± SEM of n = 5 (blood) and n = 4 (bone marrow, spleen) mice examined over at least three experiments.

particular iMoP, have the potential to localize to the nascent granuloma, we performed adoptive transfer of monocyte progenitors (CD115$^+$ CD11b$^-$) sorted from the bone marrow of $\beta$-actin gfp$^{+/-}$ mice at day 20 after infection (Fig. 8g). Again, we could identify the transferred precursors as GFP$^+$ cells at day 7 post transfer (Fig. 8h). We found that 70–80% of the transferred cells expressed CD11b after 7 days in spleen and liver. F4/80 expression was slightly higher in the spleen with around 60% than in the liver with around 40% (Figs. 8h and i). Finally, we quantified the GFP$^+$, and therefore transferred cells, in immunofluorescence stainings surrounding CD68-positive granulomas (Fig. 9a and Supplementary Fig. 6d). Around 70% of the transferred precursors were in close contact or part of BCG-containing granulomas in the liver, whereas in the spleen the distribution was more variable with around 40% of the cells being associated to granulomas (Fig. 9b). We performed a 3D reconstruction of a granuloma and the GFP$^+$ cells to further visualize their relation and the individual morphology of the transferred cells (Fig. 9c). We could hereby show that immature bone marrow cells can quickly transform and differentiate in vivo and integrate into mycobacterial granulomas.

**Transferred bone marrow precursors contribute to MGC formation in *M.tb* infected IL-13$^{tg}$ mice.** In order to further substantiate the MGC-forming potential of monocyte progenitors, we studied the fate of isolated progenitors in the lung of *M.tb* infected mice. Accordingly, we infected IL-13 over-expressing (IL-13$^{tg}$) mice with a low dose of 65 CFU *M.tb* via the aerosol route. In contrast to other mouse models, including C57BL/6 and 129S2 wild-type mice, IL-13$^{tg}$ mice develop granulomas with histological structures similar to those found in tuberculosis patients, including central necrosis and multinucleated giant cells[39]. To evaluate whether bone marrow progenitor cells develop into MGC under inflammatory conditions during experimental tuberculosis, we performed an intratracheal adoptive transfer of CD115$^+$ CD11b$^-$ Ly6C$^+$ bone marrow cells, including cMoP and iMoP, from $\beta$-actin gfp$^{+/-}$ mice 6 weeks after *M.tb* aerosol infection of IL-13$^{tg}$ mice (Fig. 9d). $\beta$-actin gfp$^{+/-}$ mice were on a C57BL/6 background and did not exhibit increased expression of IL-13. Three weeks after cell transfer, i.e., 9 weeks after infection, IL-13$^{tg}$ mice were sacrificed and lungs were analyzed by histology. In addition to GFP, we analyzed transferred cells for iNOS expression, which is a characteristic cell-intrinsic marker during MGC development[10]. Analysis of tissue slides revealed individual GFP$^+$ cells expressing iNOS (Figs. 9e1 and 9e2). Moreover, the GFP$^+$ transferred progenitors contributed to iNOS-positive cell conglomerates resembling

granulomas (Fig. 9e1). In addition, we identified transferred cells with atypical nuclear structures, including very large nuclei (Fig. 9e3) and GFP$^+$ cells with 3 nuclei (Figs. 9e4 and 9e5) in line with our definition of MGC. Thus, individual bone marrow progenitors have the potential to acquire signature MGC properties in *M.tb* infection in vivo.

## Discussion

Adaptation of the host to its microbiota requires tight local immune control, which balances containment of microorganisms and avoidance of collateral tissue damage with subsequent organ failure. In other words, a very targeted, yet overall restrained host response is the basis for efficient commensalism and thus ultimately for the benefit of the microorganism. It is an intriguing hypothesis that MGC are involved in mycobacterial latency, a particular form of host pathogen coexistence. They are reduced or missing in situations of mycobacterial dissemination, such as primary immunodeficiency syndromes or anti-TNFα treatment[10,40,41]. Yet, the specific contribution of MGC to mycobacterial immunity is still uncertain[42]. MGC may inhibit mycobacterial cell-to-cell spread[43], but may just as well promote tissue destruction due to their high expression of extracellular matrix-degrading enzymes[9]. The findings presented here suggest that the single-cell adaptation to becoming an MGC requires progenitor traits, especially proliferative activity, and commitment to the monocyte lineage. However, the progression from a dedicated monocyte progenitor to the fully differentiated MGC involves stepwise cellular reprogramming in a fashion distinct from that leading to the formation of mature mononuclear MΦ.

Diverse, and in part controversial, evidence for the role of differentiated monocytes in mycobacterial pathogenesis has been previously provided. In humans with active tuberculosis monocytes exhibit a regulatory program that predisposes them to differentiation into alternatively activated MΦ, which are relatively permissive for mycobacteria[44]. Ccr2$^{-/-}$ mice, which are deficient in circulating monocytes, develop smaller lung granulomas and release less abundant IFNγ into the draining lymph nodes after embolization with beads coupled to *M. bovis* components[45]. This may be due to impaired trafficking of antigen-presenting cells[46], since monocytes acquire DC characteristics while delivering *M.tb* to lymph nodes[47]. Moreover, type I IFN leads to a CCR2-dependent accumulation of permissive lung MΦ and neutrophils in mice, which promotes tuberculosis exacerbation[35,48]. These data are generally in line with our finding that CCR2-expressing monocyte progenitors contribute to the formation of the granuloma characteristic MGC in mycobacterial infection. Yet, CCR2

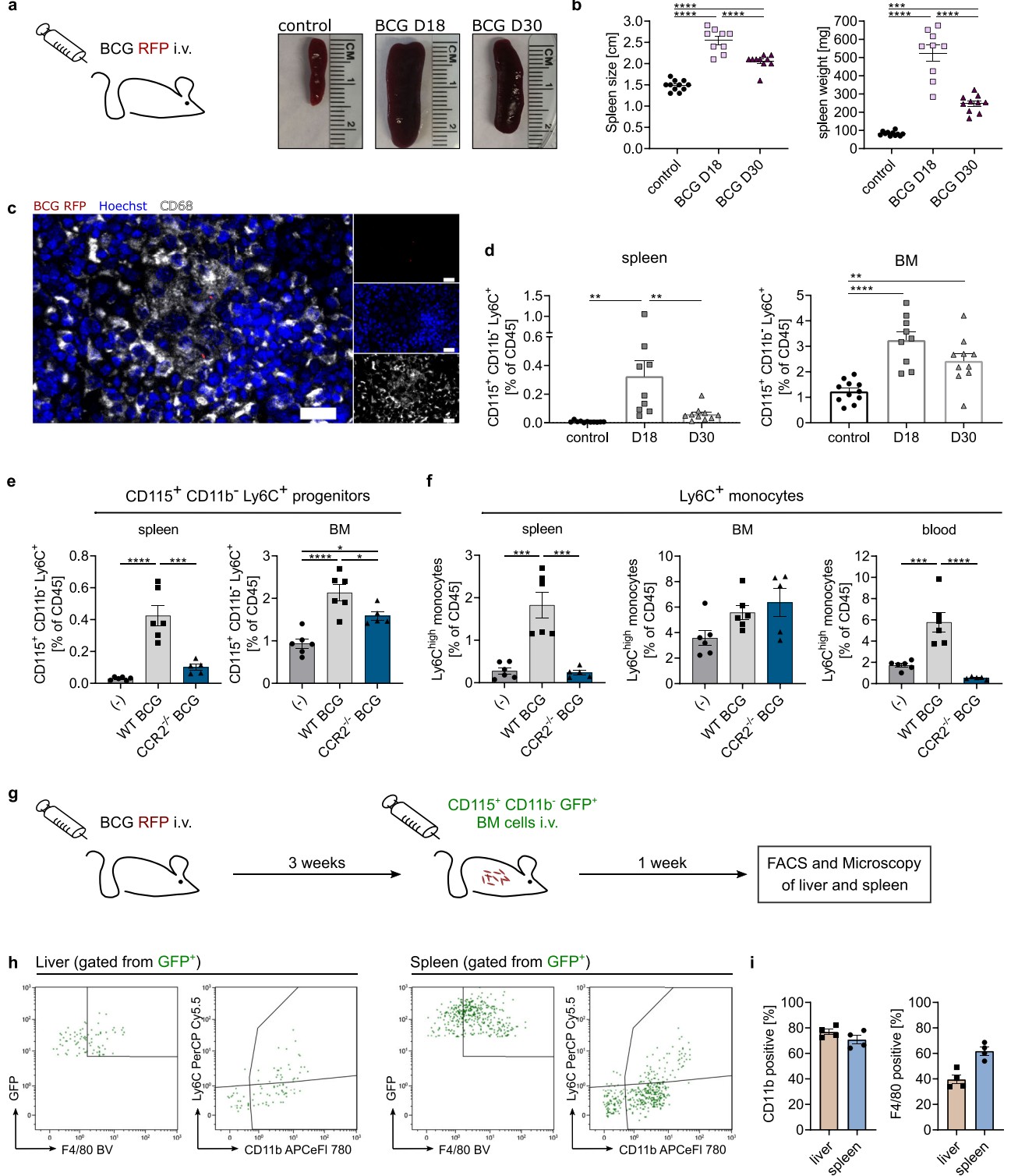

expression appeared to not be an absolute prerequisite for MGC formation, both in vitro and in vivo. It is tempting to speculate that interference of monocyte egress from the bone marrow in chronic infection triggers complex compensatory programs at the very heterogeneous host microbe-interface in the tissue.

Previously, it was described that monocytes and MΦ can be renewed independently of DCs from a committed clonogenic progenitor present in the bone marrow and spleen denominated cMoP[15]. cMoP can be found, albeit in much lower frequency than in the bone marrow, in the spleen, as an important site of

extramedullary hematopoiesis in mice[49]. In the spleen, cMoP-derived monocytes have the ability to differentiate into spleen-resident MΦ after depletion of MΦ or upon inflammation[15]. In addition, the precursor reservoir in the spleen was not empty during BCG infection of $Ccr2^{-/-}$ mice. This may be due to the reduced, but not abolished, progenitor recruitment in CCR2-deficient mice. On the other hand, it may point towards a bone marrow independent reservoir in the spleen as an alternative source of progenitor cells in $Ccr2^{-/-}$ mice to cope with the infection.

**Fig. 8 Increased progenitor frequency in spleen and bone marrow after intravenous BCG infection. a**, **b** C57BL/6 mice were infected with $1–5 \times 10^7$ CFU *M. bovis* BCG-RFP intravenously (scheme) and analyzed for splenomegaly at day 18 and 30 post infection (p.i.) compared to PBS controls. Representative images shown (**a**); quantification of spleen size and weight (**b**). Graphs show mean ± SEM of $n = 11$ (control), $n = 9$ (D18) and $n = 10$ (D30) mice examined over eight experiments. ***$p = 0.0002$, ****$p < 0.0001$ (one-way ANOVA, Tukey's MCT). **c** Immunofluorescence staining for CD68 (white) and Hoechst 33342 (blue) of spleens after BCG infection (red) as described in Fig. 8a. Scale bar: 20 μm. Channels were adjusted individually with respect to contrast and brightness. **d** FACS analysis from BCG-RFP infected and control mice (as described in Fig. 8a) for frequency of Ly6C$^+$ CD115$^+$ CD11b$^-$ cells in spleen and bone marrow (BM) at day 18 and 30 p.i. Graphs show mean ± SEM of $n = 11$ (control), $n = 9$ (D18) and $n = 10$ (D30) mice examined over eight experiments. **$p = 0.0013$ (control – D18, spleen), **$p = 0.0086$ (D18 – D30, spleen), **$p = 0.0053$ (BM), ****$p < 0.0001$ (one-way ANOVA, Tukey's MCT). **e**, **f** FACS analysis from BCG-RFP infected C57BL/6 (WT) and *Ccr2*$^{-/-}$ mice (blue) for frequency of Ly6C$^+$ CD115$^+$ CD11b$^-$ cells (**e**) and Ly6C$^{high}$ monocytes (**f**) in spleen, BM and blood at day 18 p.i. compared to PBS controls (C57BL/6) in the described infection model (Fig. 8a). Graphs show mean ± SEM of $n = 6$ (control, WT) and $n = 5$ (*Ccr2*$^{-/-}$) mice examined over two experiments (one experiment removed due to high variance in initial infection dose). *$p = 0.017$ (control − WT BCG), *$p = 0.0449$ (control − *Ccr2*$^{-/-}$ BCG), ***$p = 0.0002$ (**e**), ***$p = 0.0001$ (**f**, spleen), ***$p = 0.0004$ (**f**, blood), ****$p < 0.0001$ (one-way ANOVA, Tukey's MCT). **g** Scheme of adoptive transfer model. CD11b$^-$ CD115$^+$ cells from *β-actin-gfp*$^{+/-}$ mice were transferred into BCG-RFP ($1–5 \times 10^7$ CFU i.v.) infected WT mice 3 weeks p.i. FACS and microscopical analysis followed at day 7 post transfer. **h**, **i** FACS analysis and quantification of CD11b, Ly6C and F4/80 expression of GFP$^+$ cells (green) in spleen and liver after adoptive transfer (Fig. 8g). Depicted are mean ± SEM of $n = 4$ recipient mice examined over three experiments.

Importantly, a human cMoP analogue has been recently identified as a CLEC12A$^{high}$CD64$^{high}$ subset in umbilical cord blood and in bone marrow. Using established gating strategies, this subset is included in the conventional GMP fraction[16]. Accordingly, cMoP have become established as the most specific monocyte progenitor in bone marrow and spleen. Under homeostatic conditions, cMoP differentiate into Ly6C$^{high}$ monocytes in the bone marrow, egress and, in part, give rise to Ly6C$^{low}$ monocytes and MΦ[50]. However, it had not previously been clarified how monocyte and MΦ transformation is achieved under conditions, when these cells are most needed for immune defense, i.e., when microorganisms infect the tissue. Here, we introduce a circulating MΦ progenitor, which, according to our in vitro and in vivo differentiation analysis, is interposed between cMoP and inflammatory monocytes. This cell type, which we denominate iMoP, is best characterized by the loss of CD117. iMoP remain negative for the myeloid lineage marker CD11b, but are positive for CD115, which is why we attribute them as a monocyte/MΦ progenitor. Chong et al. established a new classification of mature bone marrow monocytes with respect to expression of CXCR4[51]. As the CXCR4$^+$ Ly6C$^+$ cells, they described, are partly CD11b$^-$, our iMoP subset might be a further subpopulation of the CXCR4$^+$ bone marrow subset. In vitro, the iMoP phenotype is, similar to the cMoP phenotype, very transient and lasts only a few hours. In accordance with this, iMoP differentiated in vivo within 14 h into CD11b$^+$ Ly6C$^+$ monocytes. The specific dedication of cMoP to its progenitor trait was based in part on strongly reduced *Ccr2* mRNA expression compared to monocytes of the bone marrow[15], which was in line with the notion that cMoP hardly egress from the bone marrow, and on their inability to take up particles as an essential functional phagocyte feature[15]. Here, we substantially extend the cellular potential of cMoP and its descendants. First, we found cMoP to express CCR2 on the protein level, and their number to be reduced in the bone marrow of CCR2-mAb treated mice. This indicates their potential to circulate and home to target tissues under certain conditions. Yet, this may occur at a very low frequency and thus evade conventional analysis. Second, we found that cMoP exhibit sensors and effector molecules involved in mycobacterial immunity. They express TLR2, a cognate receptor for mycobacterial glycolipids and lipoproteins and produce abundant TNFα in response to TLR2 ligation. In addition, a unique regulation of lipid metabolism in cMoP appears to bridge progenitor and immune cell functions. Conversion of MΦ into MGC is associated with foamy cellular transformation, as indicated by the accumulation of cholesterol and lipids in the cytosol. Our data strongly suggest that reprogramming of fatty acid and

cholesterol metabolism and efflux are critical prerequisites for MGC formation. The metabolic MGC imprint likely involves high availability of free fatty acids, either through de novo fatty acid synthesis or degradation of triacylglycerols, since both are inhibited by the general lipase inhibitor tetrahydrolipstatin (orlistat). Moreover, and probably more important in the context of our study, orlistat tightly binds and irreversibly inhibits the fatty acid synthase[28] and triacylglycerols degradation[29]. However, the cellular processes leading to the accumulation of lipids and the transformation into a foamy cell remain highly cell specific. As an example, whereas orlistat has potent activity against the accumulation of fatty acids in MGC, it increases lipid droplets in hepatic stellate cells in number and size. Under these conditions, orlistat inhibits the breakdown of cholesterol esters[29]. Accumulation of fatty acids is a clear indication for changes in the lipid handling by MGC, yet the biological role of this phenomenon is less clear. The accumulation of cholesterol and lipids in MGC may represent an enlarged reservoir, since less membrane material is needed in one slowly enlarging cell as compared to macrophages undergoing cellular fission and daughter cell formation. The inhibition of MGC formation by orlistat highlights the role of fatty acid de novo synthesis and metabolism for cellular energy production. In addition, *M.tb* uses host lipids to produce energy largely through β-oxidation, to generate biosynthetic progenitors needed for pathogenesis and to incorporate them into membrane phospholipids to maintain cytoplasmic membrane integrity[52]. Overall, host cholesterol appears to be required for proper growth and persistence of *M.tb* in vivo[53–55]. In addition, a high cholesterol content in the cell membrane supports the uptake of mycobacteria[56]. In line with this, interfering with cholesterol synthesis has been shown to reduce mycobacterial numbers both in MΦ in vitro and in experimental mouse models[57,58] and has been used as an adjunct therapy in patients with tuberculosis[59].

Furthermore, cholesterol import is a prerequisite for mycobacterial persistence in the lungs of chronically infected animals, where mycobacteria localize to MΦ[60]. It is well appreciated that *M.tb* infection induces a foamy phenotype in mononuclear MΦ, which may provide a permissive growth environment for the bacteria. Our data on increased mycobacterial loads in monocyte progenitor as compared to MC is in line with this notion, although in vitro killing assays have to be taken with some caution, given the complexity of a mixed bacterial-host cell culture (e.g., competitive metabolism, cell death etc.). Moreover, our data clearly indicate that the dependence on cholesterol for MGC transformation is independent of mycobacterial metabolism, since MBCD inhibited MGC in response to both pure TLR2 ligands and metabolically inactive mycobacteria. With regard to

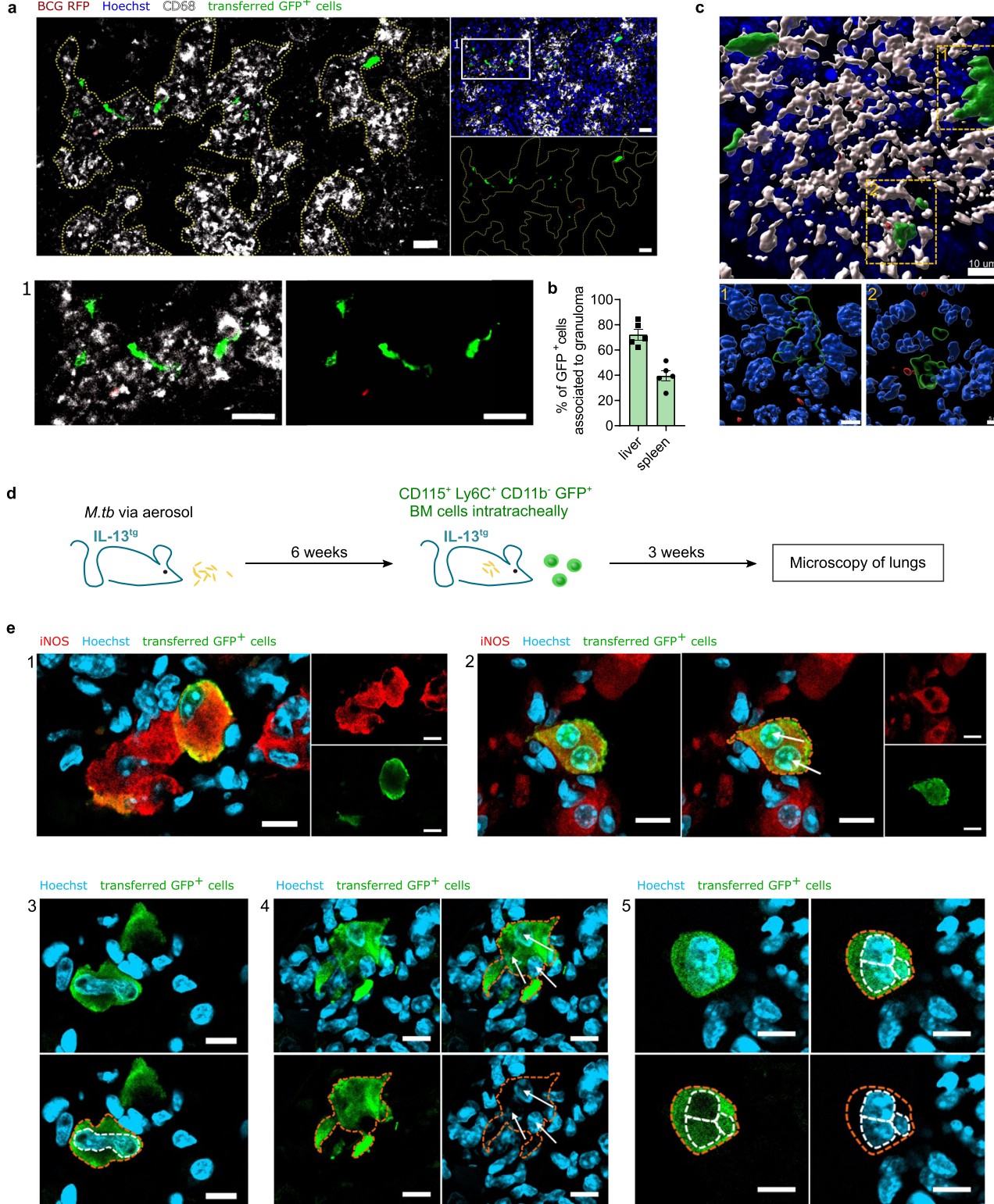

the findings of Tornack et al.[61] who detected *M.tb* DNA in peripheral blood and hematopoietic stem cells of humans during latent Tb infection, the accumulation of lipids in progenitor cells might also provide a suitable environment for mycobacteria. Moreover, it is conceivable that monocyte progenitors may serve as Trojan horses for mycobacteria contributing to Tb reactivation.

Spann et al. reported a link between the process of foam cell transformation and inflammatory response via accumulation of desmosterol[31]. Our sterol analysis of cMoP-derived MGC also

revealed an accumulation of the cholesterol precursor desmosterol. Nonetheless, TLR2-induced cMoP-derived MGC differ in the expression of e.g., *Dhcr24* and *Abca1* from the atherosclerosis-associated foam cells Spann et al.[31] described, indicating further regulatory mechanisms to play a role during the transformation process into a MGC.

The role of MGC has been a matter of considerable debate. The most accepted hypothesis, which incorporates findings in osteo-clasts, states that MGC are superior to mononuclear phagocytes

**Fig. 9 Transformation of bone marrow precursors in BCG infected wild-type mice and *M.tb* infected IL-13[tg] mice. a** Representative immunofluorescence staining of CD68-positive liver granulomas (yellow dotted line) surrounded by GFP[+] cells (green) after adoptive transfer as described in Fig. 8g. CD68 staining is shown in white, Hoechst staining in blue and BCG-RFP in red. (1) shows an enlarged section of a granuloma. Scale bar: 20 µm. Channels were adjusted individually with respect to contrast and brightness. **b** Quantification of granuloma-associated transferred cells in immunofluorescence stainings of spleen and liver after adoptive transfer as described in Fig. 8g. Depicted are mean ± SEM of $n = 5$ mice examined over four experiments. **c** 3D reconstruction of a CD68-positive (white) granuloma (scale bar: 10 µm) and detailed close-ups (1 and 2, scale bar: 5 µm) of GFP[+] cells (green) performed from immunofluorescence staining described in Fig. 9a. BCG-RFP is shown in red and nuclei are depicted in blue. **d** Scheme of adoptive transfer model. CD11b[−] CD115[+] Ly6C[+] cells isolated from *β-actin-gfp*[+/−] mice were intratracheally transferred into $n = 3$ *M.tb* infected IL-13[tg] mice 6 weeks post infection. Mice were infected via aerosol with *M.tb* (H37Rv, 65 CFU). Microscopical analysis followed 3 weeks post transfer. **e** Individual GFP[+] cells from the model described in Fig. 9d are shown. Nuclear stain (Hoechst 33342) is depicted in blue, iNOS staining in red and transferred GFP[+] cells in green (scale bars: 10 µm). Adoptive transfer and staining were performed for $n = 3$ recipient mice. 2 and 4 depict maximum intensity projections of several Z-stacks. Channels were adjusted individually with respect to contrast and brightness. The orange dotted lines represent the cell boundaries, the white dotted lines or white arrows mark the nuclei.

in the uptake and degradation of large particles due to their size and increased functional properties of the endosomes[62,63]. However, this hypothesis has been disputed recently since MGC have been found to be a less hostile environment to mycobacteria than other macrophage-like cells and rather allow for cellular persistence[10,64,65]. Our finding that dedicated monocyte progenitors are programmed by the infecting mycobacteria to progress towards cell subsets facilitating replication and persistence extends existing models of host-mycobacteria adaptation. Mycobacteria piggy-back the evolutionarily conserved antibacterial nitric oxide response to create a permissive cellular environment. At the same time, MGC locally contain the infection rather than allowing bacterial spread. MGC are relatively immobile and exhibit heightened phagocytic properties, which allow them to engulf apoptotic infected cells in the immediate vicinity. Thus, mycobacteria-driven genesis of MGC encompasses reprogramming of dedicated monocyte progenitors and represents an intriguing example of host-microbe coevolution offering mutual benefits. The hypothesis that mycobacteria have the potential to reprogram progenitor cells is supported by the observations that BCG and *M.tb* differentially impact on the transcriptional program in stem cells in vivo[66,67]. Adoptive transfer of CD115[+] CD11b[−] immature bone marrow cells yielded that these cells can migrate to *M. bovis* BCG-induced liver granulomas and acquire the morphology of granuloma-associated macrophages. As most Tb mouse models do not show typical granuloma formation in *M.tb* infection, we further analyzed the role of progenitor cells in *M.tb* infected IL-13[tg] mice which are capable of forming MGC[39]. After intratracheal transfer of CD115[+] Ly6C[+] CD11b[−] bone marrow cells, we observed cells with 3 nuclei and iNOS expression as markers for MGC formation[10]. At this stage the contribution of a T helper 2 inflammatory environment remains to be established.

In summary, we herewith introduce both cMoP and their circulating progeny iMoP as myeloid cell hybrids, which comprise both progenitor potential and immune cell functions in mycobacterial infections. The identification and characterization of dedicated, and potentially circulating MGC progenitors, opens perspectives in defining, and eventually modulating, the role of MGC in mycobacterial granulomas and beyond.

## Methods

**Mice.** All knock-out mice were on C57BL/6 genetic background and used at 6 to 10 weeks of age. Female and male mice were included in the experiments. C57BL/6J and C57BL/6N mice were purchased from Jackson Laboratories (USA) or Charles River Laboratories (Germany). *Cx3cr1*[gfp/+] mice were obtained as a kind gift from Steffen Jung (Weizmann Institute, Israel). iNOS-deficient mice (*Nos2*[tm1Lau]) and *β-actin gfp*[+/−] (C57BL/6-Tg(CAG-EGFP)131Osb/LeySopJ) were purchased from Jackson Laboratories (USA). Andreas Diefenbach (Institute of Microbiology, Infectious Diseases, and Immunology, Berlin) provided *Ccr2*[−/−] mice for infection with *M. bovis* BCG as a kind gift. Mice were bred in animal

facilities of the University of Freiburg under specific pathogen-free conditions. Mice were kept in groups of maximum of five animals at a room temperature of 21 °C (±0.5–1 °C) and a humidity of 55% (±5–8%). Food and water were available ad libitum, and day/night cycles were set to 12 h. Animal experiments under Biosafety level 2 were approved by the Regierungspräsidium Freiburg (G-19/171). For *M.tb* infections, 129S2 and C57BL/6J mice, originally bought from Charles River Laboratories (Germany), were bred at the Max-Planck-Institute for Infection Biology in Berlin. Mice between 9 and 12 weeks old were infected and maintained under Biosafety level 3. Animal experiments were approved by the State Office for Health and Social Services (Landesamt für Gesundheit und Soziales), Berlin, Germany (G040393/12). IL-13-over-expressing ([tg]) mice[68] were on a C57BL/6 genetic background and bred under specific pathogen-free conditions at the Research Centre Borstel. Infections of IL-13[tg] mice with *M.tb* were performed under Biosafety level 3 and approved by the Animal Research Ethics Board of the Ministry of Energy, Agriculture, the Environment, Nature and Digitalization Schleswig-Holstein, Kiel, Germany (approval number 3-1/19). CCR2-deficient mice for *M.tb* infection were purchased from Jackson Laboratories (USA) and bred at the animal facility at Washington University in St. Louis. The *M.tb* infection experiments of *Ccr2*[−/−] mice were in accordance with the Washington University in St. Louis Institutional Animal Care and Use Committee guidelines.

***Mycobacterium bovis* BCG.** *M. bovis* BCG-medac/Vejicur (Bacillus Calmette-Guérin) and with pSMT3-dsRed+hygromycin (Prof Zakaria Hmama, Vancouver) transformed BCG-RFP were grown to log-phase in Difco Middlebrook 7H9 broth (Becton Dickinson, BD), supplemented with 10% Middlebrook OADC Enrichment (Becton Dickinson, BD) and 0.5% Glycerol, with the addition of 50 µg/ml Hygromycin B (*Streptomyces sp.*, 400052, Merck Millipore) for BCG-RFP. BCG was heat-fixed by incubating for 30 min at 80 °C on a shaker.

***M.tb* infection of C57BL/6 and 129S2 mice and tissue processing for cell analysis.** Female C57BL/6 and 129S2 mice were co-housed one week prior to experiments. Mice were infected with low dose *Mycobacterium tuberculosis* (strain H37Rv, 200 CFU), using the Glas-Col inhalation exposure system. 24 h after infection, the amount of inhaled bacteria was controlled in five mice by plating the whole lung and counting CFU after 21 days of incubation at 37 °C. 7, 14, and 21 days after infection, blood and bone marrow from mice were analyzed by flow cytometry for progenitor frequency. Briefly, mice were lethally anesthetized with Ketamin/Xylazin i.p. and blood was readily collected from the inferior vena cava in syringes containing anti-coagulant (EDTA). Bone marrow was collected by flushing femoral bones with RPMI1640 media. Cells were resuspended in FACS buffer (1% FBS, 2 mM EDTA), blocked for unspecific binding with rat-IgG and anti-FcR antibodies and stained in the dark for 30 min at 4 °C. Details on antibodies used are presented in the section: Surface and intracellular staining. Finally, samples were fixed in 4% PFA for 30 min at room temperature (RT). Staining of blood cells was performed in whole blood. Following blocking and antibody incubation, it was transferred into fix/lyse solution (eBioscience) for 30 min at RT. Cells were acquired on the CytoFLEX S (Beckman Coulter) flow cytometer and analyzed with the FlowJo software (Treestar).

**CCR2-antibody injection in vivo.** For analysis of CCR2-dependent recruitment of iMoP, 129S2 mice were infected with a low dose of *M.tb* (H37Rv) as indicated. Starting on day 14 post infection (p.i.), mice were intraperitoneally (i.p.) injected with 20 µg of a CCR2 specific antibody (MC-21) or an isotype control (MC-67) daily. On day 14 and 19 p.i., five mice per group were euthanized and lungs and spleen plated for CFU assessment. Cells from blood and bone marrow were analyzed for frequency of cMoP and iMoP at day 19.

***M. bovis* BCG infection model.** C57BL/6 or *Ccr2*[−/−] mice were infected with ~1–5 × 10[7] CFU *M. bovis* BCG intravenously. Eighteen and thirty days after infection mice were sacrificed and spleen size and weight were measured. Spleen,

blood, and bone marrow were analyzed by flow cytometry for frequency of progenitors and other immune cells. Spleens were analyzed for granuloma formation by confocal microscopy. Briefly, mice were lethally anesthetized and blood was collected from the retroorbital plexus with capillaries containing anti-coagulant (EDTA), followed by a RBC lysis (eBioscience™ 1X RBC Lysis Buffer). Bone marrow was collected by flushing femoral bones with FACS buffer (1% FBS, 2 mM EDTA). Spleens were smashed through a 70 μm strainer and RBC lysis (eBioscience™ 1X RBC Lysis Buffer) was performed. After washing cells were resuspended in FACS buffer (1% FBS, 2 mM EDTA), blocked for unspecific binding with anti-FcR antibody and stained in the dark for 30 min at 4 °C. Details on antibodies used are presented in the section: Surface and intracellular staining. Cells were analyzed at a 3-laser flow cytometer (Gallios™, Beckman Coulter) and processed with the Kaluza software (v1.5, Beckman Coulter).

**Adoptive transfer model of _M. bovis_ BCG infected mice.** C57BL/6 mice were infected with $1–5 × 10^7$ CFU _M. bovis_ BCG intravenously. At day 20 post infection CD115$^+$ CD11b$^-$ GFP$^+$ bone marrow cells were intravenously injected. These cells were isolated from _β-actin gfp_$^{+/-}$ mice, continuously expressing GFP on their bone marrow cells except from erythrocytes. The Dynabeads™ (Thermo Fisher Scientific) purification of bone marrow cells for cell sort is described in the section: Isolation of bone marrow progenitor cells. The remaining bone marrow cells were stained with CD11b APC eFluor 780 (eBioscience) and CD115 REAlease® PE antibody (Miltenyi Biotec). After cell sort of CD115$^+$ CD11b$^-$ GFP$^+$ cells, bound CD115 REAlease® antibody was removed with a REAlease® kit (Miltenyi Biotec) following the manufacturer´s instructions. Cells were resuspended in PBS + 2% murine serum. Murine serum was obtained from donor mice and sterile filtered. Afterwards cells were injected intravenously into infected WT recipient mice ($~0.6–1.2 × 10^6$/animal). Control mice received PBS + 2% murine serum alone. Recipients were analyzed 7 days post transfer for surface marker (CD11b, Ly6C, F4/80) expression on GFP$^+$ transferred cells in blood, spleen, bone marrow, and liver by flow cytometry. Blood, spleen, and bone marrow were collected and prepared as described in the section: _M. bovis_ BCG infection model. The liver was smashed through a 100 μm strainer, followed by a RBC lysis (eBioscience™ 1X RBC Lysis Buffer). After washing cells were resuspended in FACS buffer (1% FBS, 2 mM EDTA), blocked for unspecific binding with anti-FcR antibody and stained in the dark for 30 min at 4 °C for CD11b, Ly6C, F4/80. Details on antibodies used are presented in the section: Surface and intracellular staining. Cells were analyzed at a 3-laser flow cytometer (Gallios™, Beckman Coulter) and processed with the Kaluza software (v1.5, Beckman Coulter).

**Confocal microscopy of spleen and liver sections.** Spleen and liver of infected and transferred mice were partly fixed in 4% paraformaldehyde for 4 h and then put into 20% sucrose till organs sank to the bottom of the tube. Afterwards organs were embedded in Tissue-Tek® O.C.T.™ Compound (Sakura) and shock frozen in liquid nitrogen. Slices of 8-μm thickness were cut at a cryotome and washed using washing buffer (0.1% Triton-X100, 1% BSA in 1× PBS). Cryotome sections were kept in washing buffer for 5 h at 4 °C in the dark. After removing washing buffer, slices were stained with CD68 AF647 (BioLegend), anti-GFP DyLight 488 (Rockland Immunochemicals), and Hoechst 33342 and incubated overnight at 4 °C in the dark. Samples were washed three times in washing buffer and mounted with ProLong™ Diamond Antifade Mountant (invitrogen). Before acquiring samples at the microscope, slices were kept for 24 h at 4 °C in the dark for drying. Confocal microscopy was performed on three slices of spleen and liver per animal. Between each slice at least 120 μm were discarded to analyze different parts of the organs. Pictures were taken on a LSM 710 and LSM 880 with a ×20 objective (numerical aperture 0.8) and analysis followed with ZEN 2012. In order to image whole organ sections of spleen and liver tile scans were performed with an overlay of 10% (Figs. 8c, 9a and Supplementary Fig. 6d). Brightness, contrast, and gamma correction were adjusted for ideal evaluation conditions. The picture for the 3D analysis of a granuloma was taken with a ×63 objective (numerical aperture 1.4) and analysis was performed with Imaris x64 9.5.0.

**Adoptive transfer of iMoP.** iMoP were isolated from _β-actin gfp_$^{+/-}$ mice, continuously expressing GFP on their bone marrow cells except for erythrocytes. The Dynabeads™ (Thermo Fisher Scientific) purification of bone marrow cells for cell sort is described in the section: Isolation of bone marrow progenitor cells. The remaining bone marrow cells were stained for CD11b, CD115, Ly6C, and CD117. After cell sort of CD117$^-$ Ly6C$^+$ CD115$^+$ CD11b$^-$ GFP$^+$ cells (iMoP), cells were resuspended in PBS + 2% murine serum. Murine serum was obtained from donor mice and sterile filtered. Afterwards cells were injected intravenously into WT recipient mice ($~1–3 × 10^5$/animal). Control mice received PBS + 2% murine serum alone. Recipients were analyzed 14 h post transfer for surface marker expression on GFP$^+$ transferred cells in blood, spleen, and bone marrow by flow cytometry. Blood, spleen, and bone marrow were collected and prepared as described in the section: _M. bovis_ BCG infection model. After washing cells were resuspended in FACS buffer (1% FBS, 2 mM EDTA), blocked for unspecific binding with anti-FcR antibody, and stained in the dark for 30 min at 4 °C for CD11b, Ly6C, F4/80. Details on antibodies used are presented in the section: Surface and intracellular staining. Cells were analyzed at a 3-laser flow cytometer

(Gallios™, Beckman Coulter) and processed with the Kaluza software (v1.5, Beckman Coulter).

**_M.tb_ infection of IL-13$^{tg}$ mice and progenitor transfer.** IL-13$^{tg}$ mice were infected with a low dose of 65 CFU _M.tb_ by aerogenic exposure in a Glas-Col inhalation exposure system. Six weeks post infection, enriched CD115$^+$ CD11b$^-$ Ly6C$^+$ GFP$^+$ bone marrow cells from _β-actin gfp_$^{+/-}$ mice were transferred intratracheally. Purification of bone marrow cells by magnetic bead-based cell enrichment prior to fluorescence-activated cell sorting (FACS) is described in the section: Isolation of bone marrow progenitor cells. Enriched bone marrow cells were stained with CD11b APC eFluor 780 (eBioscience), CD115 BV 421 (BioLegend) and Ly6C PerCP-Cy5.5 (BioLegend). FACS was performed on a FACS Aria Illu (BD Bioscience). $2–4 × 10^5$ CD115$^+$ CD11b$^-$ Ly6C$^+$ GFP$^+$ progenitor cells were transferred per infected IL-13$^{tg}$ mouse. Intratracheal cell transfer was achieved by aspiration of the resuspended progenitors. In short, mice were anaesthetized with isoflurane, progenitor cells resuspended in 30 μl PBS and installed in the oropharynx of the infected mice. Control mice only aspirated PBS. Recipients were analyzed 3 weeks post transfer. Lungs were paraffin-embedded for histology and microscopic slices of 6-μm thickness cut on a microtome. After air-drying slides were deparaffinized and antigen-retrieval performed in citrate buffer (10 mM citric acid, 0.05% Tween 20, pH 6.0) for 20 min at 90 °C. Blocking and immunofluorescence staining for NOS2 (Thermo Fisher Scientific), anti-GFP DyLight 488 (Rockland Immunochemicals), and Hoechst 33342 were performed similar to the description in the section: Confocal microscopy of spleen and liver sections. Confocal microscopy was performed on a LSM 880 with a ×40 objective (numerical aperture 1.2) and analysis followed with ZEN 2012. Brightness and contrast were adjusted for ideal evaluation conditions.

**Histopathological analysis of _M.tb_ infected CCR2-deficient mice.** _Ccr2_$^{-/-}$ and C57BL/6 control mice were infected with a low dose (100 CFU) of _M.tb_ (hyper-virulent strain HN878) in a Glas-Col inhalation exposure system. Lungs were harvested 60, 100, and 300 days post infection and paraffin-embedded. Paraffin sections were deparaffinized followed by an H&E staining. In short, Haematoxylin (GIL III) was added for 4 min. Slides were washed in tap water for 5 min, rinsed with 0.1% HCl and again washed in tap water. Eosin 1% was added for 1 min, followed by washing in tap water. After dehydration slides were mounted in ROTI®Histokitt (Carl Roth). Slides were scanned with a ZEISS Axio Scan.Z1 with a ×20 objective, followed by a histopathological analysis with respect to area of inflammation, foam and giant cells, and cellular composition of the inflammatory area.

**Cell culture.** After isolation, cells were cultured at 37 °C in Opti-MEM® I Reduced Serum Medium, GlutaMAX™ Supplement with 10% FBS (Gibco, Thermo Fisher Scientific) and Ciprofloxacin (5 mg/ml, Fresenius) in 48- or 96-well plates (Cell-Bind Surface, Corning). Cells were cultured with 50 ng/ml recombinant murine M-CSF from Peprotech. For stimulation, ligands or bacteria were added to the wells. Lipomannan from _M. smegmatis_ and fixed _M. tuberculosis_ were purchased from InvivoGen. BCG (MOI 20 or 100) or fixed BCG ($10^6$/ml) were used for stimulation.

**Surface and intracellular staining.** Ter119 Biotin, CD3e Biotin, CD19 Biotin, SiglecF Biotin, Sca-1 Biotin, CD127 Biotin, CD115 PE, CCR2 APC, CD11b Vio-Green anti-mouse antibodies were purchased from Miltenyi Biotec. CD16/32 purified (Fc-Block), Ly6G Biotin, CD117 Pacific Blue, CD117 BV 421, CD115 BV 421, F4/80 BV421, CD3e APC anti-mouse antibodies and Rat IgG2a Isotype BV421 were bought from BioLegend and CD4 PerCP Cy5.5, CD45 FITC, CD45 eFluor 450, CD45 PerCP Cy5.5, Ly6C PerCP Cy5.5, CD11b APC-eFluor 780, F4/80 PE, iNOS APC, Streptavidin PeCy7 and Rat IgG2a Isotype PE anti-mouse antibodies were purchased from eBioscience. Ly6G FITC anti-mouse antibody was from BD biosciences. F4/80 APC was purchased from Bio-Rad. Cells were stained in FACS buffer containing 1X Dulbecco's Phosphate Buffered Saline (Gibco, Thermo Fisher Scientific) with 1% FBS and 2 mM EDTA for 30 min at 4 °C.

For CCR2 staining cells were incubated with anti-CCR2-Antibody (MC-21)[69], or isotype control. Samples were then stained with rabbit-anti-rat secondary antibody conjugated to Alexa Fluor-488 (Thermo Fisher Scientific).

For intracellular staining, cells were fixed and permeabilized with the Fixation/Permeabilization Solution Kit (BD Bioscience) according to the manufacturer's instructions. Samples were analyzed at a 3-laser flow cytometer (Gallios™, Beckman Coulter) and processed with the Kaluza software (v1.5, Beckman Coulter). Hoechst, TNFα and filipin staining were recorded with a BD LSR Fortessa flow cytometer (Becton Dickinson). Details on FACS antibodies are provided in Supplementary Table 1.

**Isolation of bone marrow progenitors.** Bone marrow cells were isolated from tibia, femur, and humerus of mice. Tissue was removed from the respective bones, and bones were flushed with sterile FACS buffer. To obtain a single-cell suspension, cells were passed through a 70 μm cell strainer. After washing, cells were stained for 20 minutes at 4 °C with biotinylated Ter119, CD3e, CD19, SiglecF, CD127, Sca-1, Ly6G antibodies in FACS buffer. After washing in PBS containing 1% BSA, labeled cells were removed using Dynabeads™ (Thermo Fisher Scientific) according to the

manufacturer's instructions. The amount of Dynabeads™ needed was titrated. Remaining cells were stained for CD115, CD117, CD11b and Ly6C. Progenitor populations were separated using a MoFlo® Astrios™ cell sorter (Beckman Coulter).

**Isolation of blood monocytes**. Mice were anaesthetized with Ketamin/Xylazin i.p. and blood was drawn from the retro-orbital plexus. Erythrocytes were lyzed in RBC Lysis buffer solution (eBioscience™ 1X RBC Lysis Buffer). The remaining cells were washed in FACS buffer and stained for CD45, CD11b, CD115, and Ly6C as described above. Ly6C$^{high}$ and Ly6C$^{low}$ blood monocytes were isolated with a MoFlo® Astrios™ cell sorter (Beckman Coulter).

**Hemacolor staining**. Hemacolor® rapid staining for blood smears (Merck Millipore) was used to fix and stain cells. Briefly, cells were fixed in staining solution 1 for 5 seconds (Methanol), followed by staining with solutions 2 and 3 for 5 s each. Cells were washed twice in Hemacolor buffer and stored in dH$_2$O at 4 °C. Microscopy of Hemacolor staining was performed with an Axiovert 200 M Apo-Tome (Zeiss, objective: x20). For processing images, ZEN 2012 (Zeiss) was used.

**Cytospin**. Progenitor cells were resuspended in 0.5% human serum albumin (H-SA) and transferred to microscope slides by centrifugation in a cytospin slide chamber for 5 min. After air drying, slides were stained by Hemacolor staining as described above and analyzed with an Axiovert 200 M ApoTome (Zeiss, objective: ×63).

**Cell density**. Sorted cMoP were seeded in 96-well plates and stimulated with M-CSF (50 ng/ml) and fixed BCG (10$^6$/ml) or with M-CSF (50 ng/ml) alone as control. At day 6 cells were scraped, stained for flow cytometry with CD45 eFluor 450 30 min at 4 °C in the dark. After washing SPHERO™ AccuCount fluorescent particles (Spherotech) were added and samples were acquired by flow cytometry. Cell number/ml was calculated according to the manufacturer´s instructions.

**Killing assay**. cMoP, iMoP, and MC were isolated from WT mice and 4000 cells/well were plated in a 96-well plate (Corning) with M-CSF (50 ng/ml) as described above. The cells were incubated for 14 h at 37 °C and then inoculated with *M. bovis* BCG RFP (MOI 10) for 6 h. Gentamicin (50 μg/ml, Thermo Fisher) was then added to eliminate extracellular bacteria. Cells were incubated for four days at 37 °C. Then, cells were gently washed with PBS. Hundred microliters of lysis buffer (0.05% SDS in PBS) were added to each well and incubated 5 min at RT. Following cell lysis, wells were thoroughly scraped. Serial dilutions of the lysates were plated on Middlebrook 7H11-agar plates and incubated for 10–14 days at 37 °C. Colonies were counted and the bacterial concentration/ml was calculated. Cells were counted by flow cytometry with SPHERO™ AccuCount fluorescent particles (Spherotech) and bacterial concentration was referred to cell concentration per well.

**CFSE staining**. 5-(and 6)-Carboxyfluorescein diacetate succinimidyl ester (CFSE) was purchased from eBioscience (eBioscience™ CFSE, Thermo Fisher). Bone marrow progenitors were stained in a 5 μmol CFSE staining solution immediately after cell sorting for 5 min at 37 °C. Subsequently, cells were washed in cold FACS buffer and resuspended in full Opti-MEM medium and cultured with respective stimulants plus M-CSF, or M-CSF alone (control). At the indicated time points, cells were harvested by scraping, stained for CD45 and analyzed by flow cytometry. To standardize the flow cytometer settings between the different time points and replicates, Flow-Set Pro Fluorophere Flow Cytometer standardization reagent (Beckman Coulter) was used. Data were analyzed by ModFit LT™ Software (Verity Software House) and FlowJo analysis software (Version 10). The first CFSE expression peak at 5 h was used as undivided generation ($i = 1$). Proliferation index

(PI) was determined as $PI = \left(\sum_1^i i * \frac{N_i}{2^i}\right) / \left(\sum_1^i \frac{N_i}{2^i}\right)$, with $i =$ generation number

and $N =$ number of cells in generation $i$, indicating the average number of divisions responding cells have undergone[70]. $N$ and $i$ were defined with ModFit LT™ Software.

**Apoptosis assay**. cMoP and MC from WT mice were cultured as described above. As a positive control for apoptosis, progenitors cultured in M-CSF were stimulated with staurosporine from *Streptomyces sp.* (1 μg/ml, Sigma Aldrich) for 5 h at 37 °C at day 6. Samples were harvested by scraping and transferred to FACS tubes together with the supernatant. After washing, cells were resuspended in 200 μl PBS and 2 ml Ethanol was added slowly during vortexing. After fixation for 30 min on ice, cells were washed twice and stained for 30 min in DNA staining solution (20 μg/ml propidium iodide, BioLegend, and 200 μg/ml RNAse A, Thermo Fisher, Scientific, in PBS). Samples were analyzed directly by flow cytometry and the percentage of cells in subG1 phase represented the apoptotic cells[71].

**MitoSpy™ NIR DilC1(5) assay**. cMoP and MC from WT mice were cultured as described above. As a positive control for apoptosis, progenitors cultured in M-CSF were stimulated with staurosporine from *Streptomyces sp.* (1 μg/ml, Sigma Aldrich)

for 5 h at 37 °C at day 6. Samples were harvested by scraping and transferred to FACS tubes together with the supernatant. Samples were stained with 50 nM MitoSpy™ NIR DilC1(5) (BioLegend) and CD45 FITC antibody for 15 min at 37 °C. After washing cells were resuspended in FACS buffer, DAPI was shortly added before analysis by flow cytometry for dead cell exclusion. Gate for apoptotic cells, so cells with a low ΔΨm (mitochondrial potential), was defined with the staurosporine positive control.

**TNFα assay**. After isolation, bone marrow progenitors from *Cx3cr1$^{gfp/+}$* mice were cultured for 5 h as described above with appropriate stimulants and together with GolgiStop™ (BD, Germany). After scraping and washing, cells were fixed by the Fixation/Permeabilization Solution Kit (BD Bioscience) according to the manufacturer's instructions. For intracellular staining, TNFα-APC-antibody from BD Pharmigen was added and staining performed at room temperature for 20 min. After washing, samples were analyzed at a BD LSR Fortessa flow cytometer (Becton Dickinson). Unstimulated cells served as control.

**Microarray**. Sample preparation for microarray hybridization was carried out as described in the Affymetrix GeneChip WT Pico Reagent Kit User Guide (Affymetrix, Inc., Santa Clara, CA, USA).

1. Affymetrix Mouse Gene 2.1 ST (compare Fig. 3 and Supplementary Fig. 2)
   In brief, 5 ng of total RNA was used for reverse transcription to synthesize single-stranded (ss) cDNA with a T7 promoter sequence at the 5′ end. Next a 3′ adaptor was added and the ss cDNA was converted to double-stranded cDNA, which acted as a template for a pre- in vitro transcription (IVT) amplification by a low-cycle PCR. Afterwards antisense RNA (complementary RNA or cRNA) was synthesized and linearly amplified by an in vitro transcription (IVT) of the double-stranded cDNA template. 20 μg of cRNA was purified and reverse transcribed into sense-strand (ss) cDNA, wheareat unnatural dUTP residues were incorporated at a fixed ratio relative to dTTP. Purified ss cDNA was fragmented using a combination of uracil DNA glycosylase (UDG) and apurinic/apyrimidinic endonuclease 1 (APE 1) at the dUTP residues followed by terminal labeling with biotin. 3.8 μg of fragmented and labeled ss cDNA were hybridized to Affymetrix Mouse Gene 2.1 ST Array Plates. For hybridization, washing, staining, and scanning an Affymetrix GeneTitan system, controlled by the Affymetrix GeneChip Command Console software v4.2, was used.
2. Clariom S Array (compare Fig. 7)
   In brief, 5 ng of total RNA was used for reverse transcription to synthesize single-stranded (ss) cDNA with a T7 promoter sequence at the 5′ end. Next a 3′ adaptor was added and the ss cDNA was converted to double-stranded cDNA, which acted as a template for a pre- in vitro transcription (IVT) amplification by a 6 cycle PCR. Afterwards antisense RNA (complementary RNA or cRNA) was synthesized and linearly amplified by an in vitro transcription (IVT) of the double-stranded cDNA template. Twenty micrograms of cRNA was purified and reverse transcribed into double-stranded (ds) cDNA, wheareat unnatural dUTP residues were incorporated at a fixed ratio relative to dTTP. Purified ds cDNA was fragmented using a combination of uracil DNA glycosylase (UDG) and apurinic/apyrimidinic endonuclease 1 (APE 1) at the dUTP residues followed by a terminal labeling with biotin. 5.5 μg fragmented and labeled ds cDNA were hybridized to Affymetrix Clariom S mouse arrays for 16 h at 45 °C in a GeneChip hybridization oven 640. Hybridized arrays were washed and stained in an Affymetrix Fluidics Station FS450, and the fluorescent signals were measured with an Affymetrix GeneChip Scanner 3000 7G. Fluidics and scan functions were controlled by the Affymetrix GeneChip Command Console v4.1.3 software.

Sample processing was performed at an Affymetrix Service Provider and Core Facility, KFB - Center of Excellence for Fluorescent Bioanalytics (Regensburg, Germany; www.kfb-regensburg.de).

Summarized probe set signals in log2 scale were calculated by using the RMA[72] algorithm for Affymetrix Gene Array 2.1 ST or the GCCN-SST-RMA algorithm for Clariom S array with the Affymetrix GeneChip Expression Console v1.4 Software. After exporting into Microsoft Excel, average signal values, comparison fold changes and significance p values were calculated. Probe sets with a fold change above 2.0 fold and a student's $t$ test p value lower than 0.05 were considered as significantly regulated.

The principal component analysis was performed using the prcomp function in R (Version 3.2, R Core Team (2016). R: A language and environment for statistical computing. R Foundation for Statistical Computing, Vienna, Austria. URL https://www.R-project.org/.) and visualized with the 'factoextra' package.

Heatmaps show z-scores, generated with the software GENE-E (https://software.broadinstitute.org/GENE-E/). Software R i386 3.3.2 was used to create a volcano plot. The frequency of Gene-Ontology processes and the pathway analysis (PANTHER pathways) were determined with the Panther Classification System[73].

**Rapamycin and 2-DG assays**. Sorted bone marrow progenitors from *Cx3cr1$^{gfp/+}$* mice were cultured as described above. 2-Deoxy-D-glucose (2-DG) (10, 20 or

30 µmol) or Rapamycin (10 nmol) were added directly after cell sorting for 6 days. MGC were quantified based on Hemacolor staining. 2-DG was bought from Sigma Aldrich and resuspended in Dulbecco´s Phosphate Buffered Saline. Rapamycin was purchased by Enzo Life Technologies and resuspended in DMSO.

**Extracellular flux analysis.** A Seahorse XFe96 culture plate (Agilent Technologies) was coated with 22.4 µg/ml Cell-Tak™ Cell and Tissue Adhesive (Corning®) for one hour at room temperature after which the Cell-Tak was removed and the wells were washed once with 200 µl water. The plate was allowed to air dry. (Cell-Tak was diluted in water, 25 µl of dilution per well, Stock: 1.39 mg/ml). Sorted cMoP were seeded in duplicate with M-CSF (50 ng/ml) and fixed BCG ($10^6$/ml) or with M-CSF (50 ng/ml) alone as control. After incubation for 6 days, cells were analyzed by glycolysis stress test on a Seahorse XFe96 Analyzer (Agilent Technologies). Accordingly, Seahorse base media supplemented with 2 mM L-Glutamine, adjusted to pH 7.4 was used as assay medium. Injection plates were prepared with 10 mM Glucose (Port A), 1 µM Oligomycin (Port B) and 50 mM 2-DG (Port C). 2-DG, L-Glutamine Solution and Oligomycin were purchased from Sigma Aldrich. The assay protocols were designed using Wave desktop software (Version: 2.4.0.60 Agilent).

**FACS analysis of filipin stained cells.** Sorted cMoP and MC from *Cx3cr1*$^{gfp/+}$ mice were cultured in 48-well cell bind plates (corning). At day 6 cells were scraped and washed in FACS buffer (PBS + 1% FBS + 2 mM EDTA). After washing cells were fixed in 2% PFA and incubated for 20 min at 4 °C. Samples were washed in HBSS and resuspended in HBSS + Filipin (Filipin III from *Streptomyces filipinensis*, Sigma Aldrich) to a final concentration of 50 µg/ml. Samples were incubated for 60 min at RT and directly acquired at a BD LSR Fortessa flow cytometer (Becton Dickinson).

**Confocal microscopy of filipin stained cells.** Giant cell formation was induced in cMoP as described above using fixed BCG. 6 days after induction, cells were infected with BCG (MOI 1), stably expressing RFP. After 24 h WGA Alexa Fluor 488 (Thermo Fisher Scientific) was added on live cells for 10 min, then Filipin staining was performed (abcam cholesterol assay kit) according to manufacturer's instructions. Analysis was performed via 740 nm two-photon-microscopy using a ZEISS LSM 880.

**RNA extraction and RT-qPCR.** Cells were lyzed in RLT lysis buffer (Qiagen) with 1% β-mercaptoethanol. RNA was extracted with RNeasy MicroKit (Qiagen) according to the manufacturer's instructions. RNA was transcribed into cDNA by iScript™ cDNA Synthesis Kit (Bio-Rad) or by Superscript IV (Invitrogen). RT-qPCR was performed with absolute qPCR SYBR Green Mix (Thermo Fisher Scientific) in a Mastercycler® ep realplex from Eppendorf. mRNA levels were normalized to *Gapdh* as housekeeping gene. Primer sequences for *Ms4a3* were obtained from Liu et al.[13]. Further primer sequences are provided in Supplementary Table 2.

**Amplex™ red cholesterol assay Kit.** cMoP from *Cx3cr1*$^{gfp/+}$ mice were lyzed for 30 min in 0.1% Triton-X after 6 days of culture with respective stimulants. Amplex™ Red Cholesterol Assay Kit (Thermo Fisher Scientific) was performed as recommended by the manufacturer. Cells were counted by flow cytometry with SPHERO™ AccuCount fluorescent particles (Spherotech) to refer cholesterol content to the cell number.

**Cholesterol oxidase assay.** Cholesterol oxidase from *Streptomyces* (2 mU/ml, Sigma Aldrich) or PBS (vehicle) was added immediately after cell sorting to cMoP from *Cx3cr1*$^{gfp/+}$ mice. Cells were cultured as described above with appropriate stimulants. MGC was quantified by Hemacolor staining. Cell density was evaluated with SPHERO™ AccuCount fluorescent particles (Spherotech) by flow cytometry. TNFα production was measured on day 6. Five hours prior to analysis GolgiStop™ was added together with LPS (100 ng/ml, Invivogen). Staining was performed as described in the section: TNFα assay. Unstimulated cells served as control.

**MBCD assay.** Methyl-β-cyclodextrin (MBCD) was purchased from Sigma Aldrich, dissolved in distilled water and sterile filtered. cMoP from *Cx3cr1*$^{gfp/+}$ mice were cultured after cell sorting with respective stimulants. MBCD (1 mmol) or $H_2O$ (vehicle, equivalent amount for 1 mmol MBCD) were added immediately to the wells. To reverse the effect of MBCD, MBCD (1 mmol) was added to the wells together with LDL from human plasma (50 µg/ml, Athens Research and Technology) and PMA (1 µg/ml, Sigma Aldrich). As control, MBCD (1 mmol) with PMA alone was used. MGC were quantified after 6 days in culture by Hemacolor staining. Cell density was evaluated with SPHERO™ AccuCount fluorescent particles (Spherotech) by flow cytometry. TNFα production was measured as for cholesterol oxidase assay.

**Orlistat assay.** cMoP from *Cx3cr1*$^{gfp/+}$ mice were cultured under conditions described above with indicated stimulants. Orlistat (100 µmol, Cayman chemical group) or an equal amount of ethanol (control) was added directly after isolation

of cMoP. Cells were cultured for 6 days at 37 °C and MGC were quantified by Hemacolor staining or by Hoechst staining. For Hoechst staining cells were harvested and fixed by Fixation/Permeabilization Solution Kit (BD bioscience) as indicated by the manufacturer. Samples were stained with 10 µg/ml Hoechst 33342 (Thermo Fisher Scientific) for 30 min at room temperature. After washing, samples were analyzed at a BD LSR Fortessa flow cytometer (Becton Dickinson) and MGC were quantified as polyploid cells ($N > 4$) from histogram peaks. Cell density was evaluated with SPHERO™ AccuCount fluorescent particles (Spherotech) by flow cytometry. TNFα production was measured as for cholesterol oxidase assay.

**Lipid extraction and analyses using liquid chromatography mass spectrometry.** cMoP and MC from *Cx3cr1*$^{gfp/+}$ mice were isolated and cultured under conditions described above with indicated stimulants. On day 6 cells were washed in cold washing buffer (PBS + 1% lipid-free BSA), scraped and cell number was measured with a TC20 automated cell counter (Bio-Rad). Samples were centrifuged and washing buffer was discarded. Cell pellets were shock frozen in liquid nitrogen and kept at −80 °C until further processing. Samples were resuspended in 100 µl of PBS and lipids were extracted using a modified Bligh and Dyer's method[74]. Briefly, chloroform:methanol (1:2, v/v) was added and the mixture was vortexed. The samples were placed on a thermomixer and shaken at 1000 rpm for 2 h at 4 °C. Phase separation was introduced by adding chloroform and water, followed by centrifugation. Lipids in the lower organic phase were collected. The aqueous phase was re-extracted and the two organic extracts were pooled and concentrated by drying under vacuum using a Centrivap (Labconco Corporation). The dried lipid film was resuspended in 100 µl of chloroform-methanol (1:1, v/v) and spiked with internal standards (d7-cholesterol (Avanti Lipidy)) prior to LC-MS/MS analyses. Sterols were analyzed using atmospheric pressure chemical ionisation (APCI) triple quadrupole mass spectrometer (SCIEX Qtrap 6500+) with upfront liquid chromatography (Agilent Technologies). Separation of sterols was achieved using a Poroshell 120 SB-C18 column (3.0 × 50 mm, 2.7 µM, Agilent Technologies), which is a modification of the method developed by McDonald et al.[75]. For mass spectrometry analyses, the instrument was operated in the multiple reaction mode (MRM). For quantitation, area under the curve was obtained for each sterols measured and normalized to the internal standard and cell number. External calibration was made for each sterol to adjust the response factor for each lipid relative to cholesterol.

**MethoCult™ assay.** cMoP and iMoP from *Cx3cr1*$^{gfp/+}$ mice were isolated and resuspended in IMDM + 2% FBS to a cell density of $10^5$/ml. The cell suspension was mixed with MethoCult™ Medium M3434 (Stem Cell Technologies) to a cell density of $10^4$/ml. 1.1 ml of the cell suspension was distributed per culture plate. The analysis was performed in duplicates. At day 10 after culture colony formation was counted. A colony was defined as a cell formation of at least 10 cells.

**Statistics and reproducibility.** For Statistical analysis, Prism 8.0 (GraphPad software) was used. Results were considered statistically significant if p values were ≤0.05. (ns, not significant; *$p < 0.05$; **$p < 0.01$; ***$p < 0.001$, ****$p < 0.0001$). Individual statistical tests and the number of biologically independent samples are described in detail in the figure legends. In short, two-tailed unpaired student's t-test was used to compare the mean of the two groups. One- or two-way ANOVA was applied for multiple comparisons, followed by a multiple comparison analysis. The statistical analysis is based on n, indicating the number of independent biological samples or mice as described in detail in the respective figure legends. Experimental repetitions in the case of representative data were performed as follows. Phalloidin and Hoechst stainings of cMoP were performed in at least three independent experiments (Fig. 1a). Hemacolor stainings, as presented in Fig. 1d, were conducted in four independent experiments. The filipin staining in Fig. 4a depicts one representative out of three independent experiments. Hemacolor stainings in Supplementary Fig. 3b were performed of $n = 15$ and in Supplementary Fig. 3c of $n = 6$ biologically independent samples. Cytospins of bone marrow precursors were performed in three independent experiments (Fig. 6d). Granuloma stainings shown in Fig. 8c were conducted with at least $n = 5$ mice. Immunofluorescence stainings of livers after cell transfer were performed for $n = 5$ mice (Fig. 9a and Supplementary Fig. 6d). Individual 3D reconstructions of GFP⁺ transferred cells were performed for five images (Fig. 9c).

**Reporting summary.** Further information on research design is available in the Nature Research Reporting Summary linked to this article.

## Data availability
All microarray data are deposited at the National Center for Biotechnology Information Gene Expression Omnibus public data base. The accession numbers for the 'Genom-wide expression study of cMoP and monocytes from murine bone marrow compared to their differentiated descendants' (Fig. 3 and Supplementary Fig. 2) is GSE117456 and for the 'Genom-wide expression study of cMoP and iMoP from murine bone marrow' (Fig. 7) is GSE117460. Other data supporting the results can be provided by the authors upon reasonable request.

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

## Acknowledgements

We are grateful for expert technical assistance to Bernhard Kremer, Anita Imm, and Jan Bodinek-Wersing. We are indebted to the lighthouse facility of the Center for Translational Cell Research (Freiburg) and the Center for Experimental Models and Transgenic Services (Freiburg) for expert assistance with cellular and mouse analysis. In addition, we thank Lan Lu from Khader Lab for technical assistance. We are indebted to Andreas Diefenbach for providing *Ccr2*⁻/⁻ mice and to Steffen Jung for the provision of mice and advice on monocyte transfer. We thank Trude Helen Flo and Alexandre Gidon for advice on confocal microscopy. For access to the ZEISS Axio Scan.Z1 and their advice, we thank Bernhard Mauerer and Rebecca Kesselring. Furthermore, we are grateful to the fluorescene cytometry core and animal facilities in Borstel for their support. Funding to P.H.

was provided by the German Ministry of Education and Research (01EK1602B and 01EO0803) and by the German Research Council (SFB/TRR167, HE3127/9, HE3127/12 and HE3127/16). A.K.L. is supported by the MOTI-VATE postgraduate program, sponsored by the Else Kröner-Fresenius-Foundation and the University of Freiburg. F.L. is recipient of an IMM-PACT stipend (DFG 413517907) and member of the Spemann Graduate School for Biology and Medicine, University of Freiburg. Thus this work was partly supported by the Ministry for Science, Research and Arts of the State of Baden-Wuerttemberg. Funding to X.L.G. was provided by the Nanyang Technological University (Nanyang Assistant Professorship) and Ministry of Education, Singapore (MOE2017-T2-1-042).

## Author contributions

P.H. and A.K.L. designed and analyzed the experiments and wrote the manuscript. A.K.L. performed the majority of the experiments with help of K.G. and F.L. *M.tb* in vivo experiments (Figs. 6a, c, g and Supplementary Figs. 5b, d) were performed and analyzed by L.S. and A.D. (*M.tb* infection and tissue processing for cell analysis). Individual experiments were performed by F.L. (confocal microscopy of filipin staining, CCR2 expression on myeloid progenitors), J.N. (confocal microscopy of spleen and liver), C.K. (Killing assay), J.K. (PCA analysis, microscopy analysis), and A.F. (Extracellular flux analysis). X.L.G. and Y.Y.P. performed the liquid chromatography mass spectrometry. M.M. provided the CCR2-antibody and isotype control. A.T. performed initial experiments. M.S. performed the histopathological analysis of *Ccr2*⁻/⁻ mice. M.D.D. and S.A.K. performed *M.tb* infections of *Ccr2*⁻/⁻ mice and provided lung paraffin sections. A.H., C.H., and A.K.L. performed the progenitor transfers in IL-13ᵗᵍ mice. All co-authors edited the manuscript.

## Funding

## Competing interests

The authors declare no competing interests.
