## [Peer Review File · Nature Communications]

Reviewers' comments:

Reviewer #1 (Remarks to the Author):

In this work by Lösslein et al., it is shown that common monocyte progenitors (cMoP) can form MGC in vitro, which is associated to their intracellular accumulation of cholesterol and lipids. In addition, the authors show in vitro that cMoP can mount an immune response against M.tb. In mice, they describe a cell population derived from cMoP, which is the inducible monocyte progenitors (iMoP) that can form MGC and become abundant in blood and bone marrow during experimental tuberculosis.

The study of the precursors of MGC is relevant since this macrophage population contribute to the core of the granuloma and is known to constitute a safe niche for the long-term survival of the mycobacteria. Despite the pertinence of the topic, there are some weaknesses that require further improvement in order to be fit for publication in this journal.

Major concerns:

- The title does not reflect as a matter of fact the conclusions of the work with regards to the formation of MGC.
- The authors have previously demonstrated that proliferating macrophages (F4/80+ Ki67+) are increased in liver granulomas of BCG-infected mice (Herrtwich et al., 2016), but the inoculation with BCG was intraperitoneally that is not a physiologically route of infection. In my opinion, the authors should provide evidence that monocyte progenitors can reach the infected lungs and accumulate in pulmonary granulomas in the context of M. tb infection. One approach may be the use of the IL-13tg mice model (Heitmann et al., 2014) in order to improve the chance to detect presence of MGC in lung granulomas of M.tb-infected mice. Indeed, unlike the human context, the mouse model does not exhibit an accumulation of MGC in lungs. Without this evidence, the clinical relevance of the current findings becomes questionable: how can monocyte progenitors (cMoP or iMoP) reach the infected lungs, get in contact with Mtb, and form MGC? What is the relevance of the antimicrobial properties described for this cell population if it does not get in contact with Mtb?
- The experimental dose for BCG (MOI 100) and LM (4 µg/ml) used to induce MGC is very high. Do you consider that in an in vivo setting a macrophage precursor will be in contact with such a high amount of antigens or number of pathogens?
- Throughout the manuscript, the treatments to induce MGC are changed and used interchangeably between assays without providing further explanations of the reasons behind those experimental decisions (for example: sometimes LM is used, others BCG -live or killed-, Mtb -live or killed-, FSL-1, etc.). Please explain.
- Figure 1
Authors define MCG as those cells having at least three nuclei and increased cytoplasmic area. It is noticeably that MC and MDP also display an increased cytoplasmic area and several cells show 2 nuclei after LM treatment (FigS1C). Why do you consider that having 2 nuclei is not enough to support the acquisition of the MCG phenotype? Will the conclusions be different if you consider those cells having 2 nuclei as MCG? Should these two-nuclei cells be considered as a different cell group?
- Figure 2
Authors infer that cMoP exhibit antimycobacterial effector properties based on the production of TNF α and nitric oxide (figs. 2E and 2F). The conclusion on the antimycobacterial properties, which is even highlighted in the title of the manuscript, is merely speculative in the absence of a specific assay aimed to determine M.tb killing or survival. Besides, the expression of iNOS is shown only for cMoP, so it is impossible to know whether they display higher or lower amount of iNOS in comparison to other cell populations (such as MC, which in turn do not transform into MGC). I suggest the authors perform a colony-forming unit (CFU) assay to fully support the microbicidal properties of the cell population in question.

- Figure 3

The authors link the lipid metabolism to MGC formation based on genes that are up-regulated in LM-stimulated cMoP vs LM-stimulated MC, but they failed to explain the fact that those genes are downregulated in cMoP upon LM-treatment compared to untreated cells. According to the presented results, it seems that the lipid metabolism is more related to cell proliferation than to MGC formation (Fig 3G). Please explain.

- Figure 4

- In order to link the cholesterol accumulation to MGC formation, the authors should show that other cell populations, which do not transform into MGC (i.e., MC), could not accumulate cholesterol in response to BCG-RFP.
- Fig 4D provides a direct link between cholesterol accumulation and MGC formation, but several important validations should be shown, such as: 1) Is cholesterol accumulation reduced after cholesterol oxidase treatment? 2) Is cell viability affected after being infected with BCG (MOI 20) and treated with cholesterol oxidase by 6 days? The same is needed for MBCD treatment. For orlistat, this information is shown (4G and I).

- Figure S4

Why are the Seahorse analyses performed using MGC from bone marrow-derived macrophages instead from cMoP? The comparison of ECAR and OCR between cMoP stimulated with LM vs MC stimulated with LM would be more informative.

- Figure 5

- The capacity of iMoP to generate MGC should be compared also with MDP (fig. 5E) because this population has been shown to display intermediate potential to produce MGC (Fig 1).
- In page 12 (line 315) the authors state "To further analyze the recruitment of iMoP to the site of infection", but the recruitment of iMoP to the site of infection is not explored within the manuscript.
- The results indicating an inhibitory effect of BAL on giant cell formation are quite preliminary to be presented in the results section. Maybe they can be commented in discussion.

Minor comments

Figure 1F: it is difficult to differentiate among the different levels of the gray color, please order the labels in the same order as they are shown (i.e. Lin⁻ c-kit⁺, MDP, cMoP and MC).

Figure 2: It is not clear why LM treatment is used to evaluate proliferation (fig. 2B) and heat-fixed M.tb to evaluate apoptosis (fig. 2C). Please explain.

Reviewer #2 (Remarks to the Author):

In their manuscript titled "Monocyte progenitors are antimycobacterial effector cells" Henneke and colleagues explore the myeloid origins of multinucleated giant cells (MGCs), enigmatic cells that arise in multiple granulomatous disorders including tuberculosis. The cellular origins and signaling processes that drive MGC development in response to inflammatory stimuli remain largely unknown and in some respects quite controversial, particularly in terms of whether or not these cells arise through cell fusion events or through defects in DNA damage response and cell cycle defects. Building on their previous work linking MGC formation to Toll-like receptor signaling-induced DNA damage, Henneke and colleagues identify a population of monocyte progenitors from mouse bone marrow that has an increased potential to produce MGCs in vitro in response to mycobacterial stimulation plus CSF-1 compared to the capacity of other myeloid cells. These progenitors give rise to an intermediate population that can be detected in the blood of Mycobacterium tuberculosis-infected mice, suggesting that it may be a physiologically relevant MGC precursor. Through a series of in vitro experiments, the authors implicate lipid uptake, cholesterol and fatty acid metabolism, and mTOR signaling in MGC biogenesis. The authors employ a range of stimulatory conditions, including several TLR2 ligands, and use two inbred mouse strains to test the robustness of their findings. This work constitutes a significant advance in our

understanding of MGC formation and should be appeal to readers interested in macrophage biology, tuberculosis, and both infectious and non-infectious granulomatous disorders.

While the authors successfully investigate the potential of several myeloid populations to produce MGCs *in vitro*, it remains unclear whether or not the inducible MoPs indeed constitute the chief progenitor population in the context of a granulomatous response *in vivo*. Furthermore, the reasons why these cells are particularly adept at producing MGCs are unclear. The authors also implicate lipid and cholesterol metabolism and mTOR signaling in MGC biogenesis. While interesting, these observations raise several questions regarding how specific these processes are for MGC development since these are pathways that are broadly engaged in response to mycobacterial infection and have been previously implicated in the development of other highly differentiated macrophage populations in TB granulomas - foamy macrophages (lipid uptake and cholesterol metabolism) and epithelioid macrophages (mTOR). While the authors identify a candidate myeloid progenitor for MGCs and delineate some signaling pathways that stimulate MGC development *in vitro*, identifying the physiological MGC progenitor(s) and untangling the differentiation programs between MGCs, epithelioid macrophages and foam cells would substantially strengthen the manuscript.

Major concerns:

1. The physiological MGC progenitor remains unclear.

Are cMoPs physiological sources of MGCs *in vivo*? The authors establish that cMoPs are superior to other myeloid subsets in terms of producing MGCs *in vitro* and show that a developmental derivative of these cells (i.e., the iMoP) circulates in the blood of *M. tuberculosis*-infected mice. What are the relative frequencies of iMoPs and monocyte populations in the blood? While other myeloid progenitors might have lower MGC potential on a per-cell basis, it is possible that if they were more abundant or mobilized more efficiently during an infection, that they might be a more relevant source of MGCs.

2. Is CCR2 signaling required for MGC formation *in vivo*?

Anti-CCR2 reduced the number of iMoPs in the blood. Does this treatment also impair MGC development *in vivo*?

3. The attributes that make cMoPs better at producing MGCs than the other myeloid populations tested remain unclear. The authors show that cMoPs retain greater proliferative capacity than monocytes in response to TLR2 stimulation. Given that their model of MGC formation involves mitotic defects, could this be part of the answer? Given that the authors have previously implicated TLR-dependent nitric oxide production in the DNA damage that is critical to MGC development, do the different myeloid precursors differ in their capacity to produce NO?

4. The roles of lipid uptake and cholesterol metabolism in MGC versus foamy macrophage and the role of mTOR in MGC versus epithelioid macrophage development are unclear. How does MGC development relate to foamy and epithelioid macrophage development? Do the proliferative capacity of the cells and/or does the presence of other stimulatory factors influence the developmental trajectories?

5. The claim that "monocyte progenitors are specific effector cells in mycobacterial immunity" (line 38) seems unwarranted. And so does "This spurs the hypothesis that iMoP are involved in the defense against mycobacteria" (lines 354-355). In the first place, the authors do not test the antimycobacterial capacity of cMoPs, and secondly, this hypothesis clashes with their previous findings showing that MGCs are less microbicidal against mycobacteria than their macrophage counterparts. Furthermore, addressing the microbicidal capacity of these myeloid populations seems like a distraction from the main goal of this manuscript of understanding MGC development.

Minor concerns:

1. The title of the manuscript does not accurately reflect the primary focus of the paper. The manuscript seems more about MGC development than about the antimycobacterial capacity of MGCs.

2. The introduction and discussion sections talk about the co-evolution of pathogenic mycobacteria and humans and allude to the possibility that MGCs might serve as a growth permissive niche for Mtb and may therefore contribute to latent TB. These statements detract from the main thrust of the paper: understanding how MGCs develop. Given that MGCs are observed in active tuberculosis and soon after mycobacterial stimulation *in vitro*, it might be more appropriate to focus the text on MGC development.

Point-by-point response to the reviewers' comments:

Reviewer #1:

The study of the precursors of MGC is relevant since this macrophage population contribute to the core of the granuloma and is known to constitute a safe niche for the long-term survival of the mycobacteria. Despite the pertinence of the topic, there are some weaknesses that require further improvement in order to be fit for publication in this journal.

Major concerns

1.1. *The title does not reflect as a matter of fact the conclusions of the work with regards to the formation of MGC.*

In accordance with both of the reviewers' requests we have modified the title to **"Monocyte progenitors give rise to multinucleated giant cells"**.

Furthermore, we have replaced the term effector cell throughout the manuscript.

1.2. *The authors have previously demonstrated that proliferating macrophages (F4/80+ Ki67+) are increased in liver granulomas of BCG-infected mice (Herrtwich et al., 2016), but the inoculation with BCG was intraperitoneally that is not a physiologically route of infection. In my opinion, the authors should provide evidence that monocyte progenitors can reach the infected lungs and accumulate in pulmonary granulomas in the context of M. tb infection. One approach may be the use of the IL-13tg mice model (Heitmann et al., 2014) in order to improve the chance to detect presence of MGC in lung granulomas of M.tb-infected mice. Indeed, unlike the human context, the mouse model does not exhibit an accumulation of MGC in lungs. Without this evidence, the clinical relevance of the current findings becomes questionable: how can monocyte progenitors (cMoP or iMoP) reach the infected lungs, get in contact with Mtb, and form MGC? What is the relevance of the antimicrobial properties described for this cell population if it does not get in contact with Mtb?*

Thank you for these important considerations. Similar to this reviewer we are intrigued by the question, whether progenitor cells are also a natural source of MGC and can contribute the granuloma in mice. Unfortunately, a really satisfactory mouse model, where mycobacterial infections lead to the formation of "typical" necrotic granulomas with frequent giant cells is not available. As an example, the suggested IL-13tg mouse model has merits, but it is prone to effects beyond mycobacterial immunopathology.

Overexpression of IL13 in the lung alone has been shown to lead to spontaneous inflammation and to DNA damage, which constitutes a major stimulus for giant cell formation (Chapman et al. 2014. *Mutat Res* 769: 100-7 and Herrtwich L. et al. 2016. *Cell* 167: 1264-1280). The alternative C3HeB/FeJ mice, which form granulomas, are not suitable for adoptive transfers of cells isolated from β -Actin *gfp*^{+/-}-mice, which are of C57BL/6 genetic background (Driver et al. 2012. *Antimicrob Agents Chemother* 56: 3181-95).

In order to still address whether progenitors can turn into granuloma macrophages, we made use of a model, where intravenous infection of BCG leads to the formation of CD68-positive granulomas in spleen and liver (**new figs. 7 and S6**, lines 434-464). Next, we performed adoptive transfer of CD115⁺ CD11b⁻ mononuclear phagocyte progenitor cells, isolated from β -Actin *gfp*^{+/-}-mice. Notably, this is a very laborious endeavor. As shown in figure 7H, these cells integrate as macrophage-like cells into BCG-containing granulomas in the liver (**new fig. 7H**). To further illustrate the size of the transferred cells and their relation to the granulomas we performed a 3D reconstruction (**new fig. 7J**).

Accordingly, although our experiments cannot solve all of the questions related to species-specific particularities in mycobacterial granuloma formation, which have been explored for decades, our adoptive transfer data provide novel evidence on defined monocyte progenitors as specific mycobacterial granuloma macrophages. Notably, in humans, pulmonary tuberculosis is not the only mycobacterial disease course. Quite in contrast, the relative proportion of extrapulmonary Tb among all Tb cases has nearly tripled in the last 50 years. It is well established that mycobacteria can also spread via the blood stream and induce granuloma formation in different organs, with the liver constituting a major site (e.g. Hickey et al. 2015. *BMC Infect Dis.* 15, 209). Thus, we firmly believe that – despite all of the limitations – i.v. BCG liver infection has merits as a mycobacterial disease model.

1.3. *The experimental dose for BCG (MOI 100) and LM (4 µg/ml) used to induce MGC is very high. Do you consider that in an in vivo setting a macrophage precursor will be in contact with such a high amount of antigens or number of pathogens?*

We agree with the reviewer that an MOI of 100 is a high infection dose *in vitro*. It seems notable, however that 11 weeks after infection of cynomolgus macaques up to 10⁵ bacteria per granuloma could be found (Gideon et al. 2015. *PLoS Pathog.* 11:e1004603), thus indicating that - *in vivo*- macrophages may be exposed to a high bacterial burden. However, in order to substantiate our findings we have performed additional experiments with an MOI of 20, which robustly induced MGC formation in progenitor cells (**fig. 1, new figs. S1A, S1B**,

S1C). Furthermore, we added the bacteria directly after progenitor isolation. The monocyte progenitors (2000 cells per well at time 0) then start to proliferate, leading to a further decrease of the MOI with time.

Lipomannan was titrated for every lot number. We always found 4 µg/ml to lead to the best induction of MGC (except for one batch, fig. 3E). In the course of our study, production of lipomannan was discontinued by the manufacturer (Invivogen) and an extensive, worldwide search for highly pure and biologically active lipomannan remained without success.

1.4. *Throughout the manuscript, the treatments to induce MGC are changed and used interchangeably between assays without providing further explanations of the reasons behind those experimental decisions (for example: sometimes LM is used, others BCG -live or killed-, Mtb -live or killed-, FSL-1, etc.). Please explain.*

We agree that the use of different stimulants might need further explanation. As described above Lipomannan of *M. smegmatis* was discontinued by the manufacturer during our study. Therefore, we used *M. bovis* BCG and *M. tuberculosis* as alternative stimulants and repeated most of our experiments, which we had performed before with Lipomannan to show the same effects on our progenitor populations for bacterial pathogens.

However, in order to simplify the ligands used as much as possible, we have replaced the lipoprotein FSL-1, which we used as an alternative TLR2 ligand in the Seahorse assay in the previous version of the manuscript, by *M. bovis* BCG (**new figs. S3A, S3B**).

1.5. *Figure 1: Authors define MCG as those cells having at least three nuclei and increased cytoplasmic area. It is noticeably that MC and MDP also display an increased cytoplasmic area and several cells show 2 nuclei after LM treatment (FigS1C). Why do you consider that having 2 nuclei is not enough to support the acquisition of the MCG phenotype? Will the conclusions be different if you consider those cells having 2 nuclei as MCG? Should these two-nuclei cells be considered as a different cell group?*

We agree with the reviewer that the population of cells with two nuclei may be heterogeneous, i.e. comprise cells that are “almost MGC”. However, since M-CSF alone induces cell proliferation, which is decreased during MGC transformation (Gharun et al. 2017. *EMBO Rep.* 18: 2144-2159 and Herrtwich L. et al. 2016. *Cell.* 167: 1264-1280), the cells with two nuclei are likely to differ in character between control and MGC inducing conditions. Accordingly, cells with two nuclei are found in high frequency in our M-CSF controls *in vitro*. For cMoP this occurred even more often than in the stimulated conditions

(**new fig. S1D**). Accordingly, cells with two nuclei cannot be used as parameters for the MGC transformation process *in vitro* (lines 130-133). However, the *in vivo* experiments (adoptive transfer) revealed that transferred progenitors develop into particularly large granuloma macrophages, which are somewhat reminiscent to MGC (albeit true MGC do not develop in this model, as discussed above).

1.6. *Figure 2: Authors infer that cMoP exhibit antimycobacterial effector properties based on the production of TNF α and nitric oxide (figs. 2E and 2F). The conclusion on the antimycobacterial properties, which is even highlighted in the title of the manuscript, is merely speculative in the absence of a specific assay aimed to determine M.tb killing or survival. Besides, the expression of iNOS is shown only for cMoP, so it is impossible to know whether they display higher or lower amount of iNOS in comparison to other cell populations (such as MC, which in turn do not transform into MGC). I suggest the authors perform a colony-forming unit (CFU) assay to fully support the microbicidal properties of the cell population in question.*

iNOS expression is a key factor in MGC formation, as shown here and in our previous study. Yet, the formation of MGC is a complex cellular transformation process effecting metabolism, DNA damage and cytokine production. However, iNOS expression alone, does not directly determine the transformation process. Moreover, as we have previously shown reactive oxygen species can partially compensate for iNOS-deficiency (Gharun et al. 2017. *EMBO Rep* 18: 2144-2159). We have carefully chosen the wording to highlight that cMoP show sufficient iNOS upregulation, which is intriguing given their progenitor status.

In order to experimentally meet the reviewer's questions on the antimycobacterial activity of i/cMoP, we have performed killing assays, and found more living BCG in MC as compared to cMoP and iMoP (**new fig. S5F**). This data suggest an increased antimycobacterial activity in iMoP progeny. However, it seems important to note that *in vitro* killing assays have several limitations, since cell death, induced macrophage transformation processes, and bacterial proliferation all go in parallel in a cell type specific fashion. The release of bacteria into the media, which is then not reflected by conventional killing assays, is another shortcoming. Accordingly, we have discussed this issue in more detail in the new version of the manuscript (lines 575-579).

1.7. *Figure 3: The authors link the lipid metabolism to MGC formation based on genes that are up-regulated in LM-stimulated cMoP vs LM-stimulated MC, but they failed to explain the fact that those genes are downregulated in cMoP upon LM-treatment compared to untreated*

cells. According to the presented results, it seems that the lipid metabolism is more related to cell proliferation than to MGC formation (Fig 3G). Please explain.

We fully agree with the reviewer that cholesterol and lipid metabolism are probably connected to proliferation, since lipids are needed for the formation of new cell membranes. However, our data clearly connect the availability of cholesterol and MGC induction (fig. 4F). As an example, MGC formation is inhibited by cholesterol extraction (methyl- β -cyclodextrin) without impacting on the inflammatory response of the cells.

In order to further explore the specific lipid metabolic traits of cMoP we have added substantially new data by performing a sterol analysis with our new collaborator Dr. Xue Li GUAN (Singapore), which show that desmosterol, a cholesterol precursor, accumulates in cMoP stimulated with BCG (**new figs. 4K and S4J**), (lines 296-317). In addition, we analyzed key enzymes of the Bloch pathway for cholesterol biosynthesis at day 6, when MGC have already formed. We found them to be significantly increased in BCG-stimulated cMoP when compared to MC (**new figs. 4L and S4K**).

1.8. *Figure 4: In order to link the cholesterol accumulation to MGC formation, the authors should show that other cell populations, which do not transform into MGC (i.e., MC), could not accumulate cholesterol in response to BCG-RFP.*

Overall, accumulation of cholesterol is not an exclusive trait of monocyte progenitors in response to mycobacteria. However, cMoP differ from mature monocytes in that they accumulated cholesterol faster, i.e. within 48 hours (**new fig. 4B**), (lines 242-245). Moreover, cMoP and MC qualitatively differ with respect to the accumulated cholesterol precursors. In addition, key enzymes of the Bloch pathway were significantly upregulated in cMoP at day 6 compared to MC, also indicating a different regulation of lipid metabolism (**new figs. 4L, S4K**). However, the transformation into MGC involves complex metabolic events, i.e. it cannot be explained by availability of one metabolite, e.g. cholesterol. Additionally, MGC formation can be inhibited by blocking the fatty acid synthase with orlistat. Accordingly, MGC transformation appears to be intertwined with the regulation of lipid and cholesterol metabolism and the profile of accumulated metabolites.

1.9. *Fig. 4D provides a direct link between cholesterol accumulation and MGC formation, but several important validations should be shown, such as: 1) is cholesterol accumulation reduced after cholesterol oxidase treatment? 2) Is cell viability affected after being infected with BCG (MOI 20) and treated with cholesterol oxidase by 6 days? The same is needed for MBCD treatment. For orlistat, this information is shown (4G and I).*

We fully agree with the reviewer that further quality controls are useful to strengthen the data. Accordingly, we incorporated a host of additional control conditions (**new figs. S4B, C, D, E, F, G**). Cholesterol oxidase treatment did not impact on cell viability. However, cholesterol oxidase catalyzes the transformation of cholesterol into cholest-4-en-3-one and therefore changes the cholesterol molecule and does not extract it from the cell. Instead, we measured filipin staining after MBCD treatment, which indeed extracts cholesterol, and found a reduction of cholesterol accumulation (**new fig. S4I**), (lines 285-294).

1.10. *Figure S4: Why are the Seahorse analyses performed using MGC from bone marrow-derived macrophages instead from cMoP? The comparison of ECAR and OCR between cMoP stimulated with LM vs MC stimulated with LM would be more informative.*

It seems important to note that the metabolic analysis in a Seahorse analyzer is hampered by several obstacles. First, the required cell culture plates do not provide the surface coating, which we have used to optimize our highly reproducible MGC protocol. One consequence is an alteration in MGC transformation, which is particularly notable for progenitors, whereas BMDMs are less affected. As we cannot normalize our results in a Seahorse assay, we rather underestimate effects in progenitor cells between stimulated and control cells. Finally, we also performed the assay with cMoP, stimulated with BCG, and found significant differences compared to control cMoP (**new figs. S3A, S3B**). A comparison to MC is still not feasible in Seahorse assay as there is no possibility to normalize for the substantially decreased proliferation rate in MC as compared to cMoP. We fully agree that MGC derived from BMDMs are conceptually not identical to cMoP derived MGC.

1.11. *Figure 5: The capacity of iMoP to generate MGC should be compared also with MDP (Fig. 5E) because this population has been shown to display intermediate potential to produce MGC (Fig. 1).*

We acknowledge that the comparison between iMoP and MDP with respect to their MGC forming properties, was not analysed in detail in the first version of the manuscript. We found that both cell types behave similar with respect to MGC per cm². However, MDP are substantially less specific with respect to MGC formation (% of total cells) and instead proliferate more into mononuclear cells (**new fig. S5A**), (lines 356-358).

1.12. *In page 12 (line 315) the authors state “To further analyze the recruitment of iMoP to the site of infection”, but the recruitment of iMoP to the site of infection is not explored within the manuscript.*

In full accordance with the reviewer’s concerns we have changed the phrase to “To further analyze the recruitment of iMoP into the blood stream...”. As described above with our new experiments (**fig. 7**) we can now indeed show the migration of transferred progenitors to the CD68-positive liver granulomas.

1.13. *The results indicating an inhibitory effect of BAL on giant cell formation are quite preliminary to be presented in the results section. Maybe they can be commented in discussion.*

We agree with the reviewer and commented on the data in the discussion (lines 499-503).

Minor comments

1.14. *Figure 1F: it is difficult to differentiate among the different levels of the gray color, please order the labels in the same order as they are shown (i.e. Lin- c-kit+, MDP, cMoP and MC).*

We have adjusted the color schemes and have always used the same order.

1.15. *Figure 2: It is not clear why LM treatment is used to evaluate proliferation (fig. 2B) and heat-fixed M.tb to evaluate apoptosis (fig. 2C). Please explain.*

As described above, lipomannan was no longer available at one point of the study. In order to support our results, we have additionally measured proliferation for BCG- stimulated cMoP (**new fig. 2C**). To assess apoptosis, we have additionally measured the mitochondrial membrane potential in BCG stimulated cells (**new fig. S3C**), (lines 146-148).

Reviewer #2

Through a series of in vitro experiments, the authors implicate lipid uptake, cholesterol and fatty acid metabolism, and mTOR signaling in MGC biogenesis. The authors employ a range of stimulatory conditions, including several TLR2 ligands, and use two inbred mouse strains

to test the robustness of their findings. This work constitutes a significant advance in our understanding of MGC formation and should be appeal to readers interested in macrophage biology, tuberculosis, and both infectious and non-infectious granulomatous disorders.

While the authors successfully investigate the potential of several myeloid populations to produce MGCs in vitro, it remains unclear whether or not the inducible MoPs indeed constitute the chief progenitor population in the context of a granulomatous response in vivo. Furthermore, the reasons why these cells are particularly adept at producing MGCs are unclear. The authors also implicate lipid and cholesterol metabolism and mTOR signaling in MGC biogenesis. While interesting, these observations raise several questions regarding how specific these processes are for MGC development since these are pathways that are broadly engaged in response to mycobacterial infection and have been previously implicated in the development of other highly differentiated macrophage populations in TB granulomas - foamy macrophages (lipid uptake and cholesterol metabolism) and epithelioid macrophages (mTOR). While the authors identify a candidate myeloid progenitor for MGCs and delineate some signaling pathways that stimulate MGC development in vitro, identifying the physiological MGC progenitor(s) and untangling the differentiation programs between MGCs, epithelioid macrophages and foam cells would substantially strengthen the manuscript.

Major concerns:

2.1. *The physiological MGC progenitor remains unclear. Are cMoPs physiological sources of MGCs in vivo? The authors establish that cMoPs are superior to other myeloid subsets in terms of producing MGCs in vitro and show that a developmental derivative of these cells (i.e., the iMoP) circulates in the blood of M. tuberculosis-infected mice. What are the relative frequencies of iMoPs and monocyte populations in the blood? While other myeloid progenitors might have lower MGC potential on a per-cell basis, it is possible that if they were more abundant or mobilized more efficiently during an infection, that they might be a more relevant source of MGCs.*

We thank the reviewer for raising these important questions. To further explore the role of progenitor cells *in vivo*, we have used an intravenous *M. bovis* BCG infection model, leading to the formation of CD68-positive granulomas in spleen and liver. The model is characterized in **figs. 7 A-D**. Similar to the situation in experimental tuberculosis, we observed an increase of CD115⁺ Ly6C⁺ CD11b⁻ precursors in spleen and bone marrow in this model. To analyze, if our cells of interest can migrate to granulomas we performed an adoptive transfer of CD115⁺ CD11b⁻ cells from the bone marrow of β -Actin *gfp*^{+/+} mice. In full accordance with the

proposed model, we found GFP-positive cells localized to liver granulomas 7 days post transfer (**new figs. 7E-I**). These observations show that progenitor cells can transform into mycobacterial granuloma macrophages *in vivo* (lines 432-464).

As already indicated by the reviewer we cannot exclude that also monocytes are recruited to the side of infection. The frequency of cells with a monocyte immunophenotype in the blood stream is higher than that of iMoP. However, as shown in new **figs. 6G and 6H** iMoP rapidly upregulate CD11b *in vivo*, also in steady state (lines 422-426). Accordingly, they are immunophenotypically “monocytes”, however, they retain their potential as MGC precursors. Thus it seems likely that considerable heterogeneity exists in the circulating population of monocytes. In line with this, a fraction of Ly6C^{hi} monocytes will transform into regulatory (Ly6C^{low}) monocytes while still in the circulation. Interestingly, we found that there was no significant increase of Ly6C⁺ monocytes in the bone marrow in BCG infection (**new fig. S6B**) in contrast to the substantial effect in progenitors (**new fig. 7D**).

2.2. *Is CCR2 signaling required for MGC formation in vivo? Anti-CCR2 reduced the number of iMoPs in the blood. Does this treatment also impair MGC development in vivo?*

As outlined above generally accepted mouse models for the specific formation of mycobacterial necrotic granulomas are not available. Moreover, CCR2 is important but not specific for iMoP, since it is involved in recruitment of monocytes as well. Thus, any effect of CCR2-deficiency cannot be directly linked to progenitors. However, previously CCR2 deficient mice have been found to exhibit an impaired resistance to tuberculosis (Antonelli et al. 2010. *J Clin Invest* 120: 1674-82) and delayed granuloma formation (Scott and Flynn. 2002. *Infect Immun* 70: 5946-54). We have added this information to the discussion.

2.3. *The attributes that make cMoPs better at producing MGCs than the other myeloid populations tested remain unclear. The authors show that cMoPs retain greater proliferative capacity than monocytes in response to TLR2 stimulation. Given that their model of MGC formation involves mitotic defects, could this be part of the answer? Given that the authors have previously implicated TLR-dependent nitric oxide production in the DNA damage that is critical to MGC development, do the different myeloid precursors differ in their capacity to produce NO?*

We appreciate that we did not sufficiently explain our model on why c/iMoP have particular MGC forming properties. In full accordance with the reviewer’s suggestion we believe that their proliferative capacity, which exceeds that of monocytes, contributes to MGC formation.

However, as outlined above, the formation of MGC is a complex cellular transformation process effecting metabolism, DNA damage and cytokine production. iNOS expression is just one – albeit very important – factor in this process. iNOS expression alone does not directly determine the extent of the transformation process. Yet in view of the hitherto restricted role of cMoP as progenitors it seems very notable that they respond with a robust iNOS and TNF α response to mycobacteria (**new fig. 5G**). The comparison of transcriptional GO-Terms in TLR2-activated cMoP and mature macrophages revealed overrepresentation of ‘mitotic DNA replication’, ‘regulation of mitotic spindle checkpoint’ and ‘DNA repair’ in cMoP’ (data not shown). Thus, it indicates a higher proliferative activity and also mitotic changes as suggested by the reviewer. This would be in accordance with our former observations (Gharun et al. 2017. *EMBO Rep* 18: 2144-2159 and Herrtwich et al. 2016. *Cell* 167: 1264-1280). Together, we are convinced that the combination of proliferative activity, mitotic defects, iNOS, and sustained lipid metabolism all contribute to MGC transformation in c/iMoP. We have improved the language referring to this concept.

2.4. *The roles of lipid uptake and cholesterol metabolism in MGC versus foamy macrophage and the role of mTOR in MGC versus epithelioid macrophage development are unclear. How does MGC development relate to foamy and epithelioid macrophage development? Do the proliferative capacity of the cells and/or does the presence of other stimulatory factors influence the developmental trajectories?*

Currently, it is not clear how distinct macrophage types in the granuloma develop, and if the borders between these cells are fixed or flexible. We define our cells as MGC because they have at least 3 nuclei. Still, the metabolic changes might be overlapping with those in foam cells. For example, the accumulation of desmosterol was also reported for atherosclerotic foam cells (Spann et al. 2012 *Cell* **151**: 138-52). A key finding in this manuscript is that an immature precursor cell is mobilized to develop into a specialized granuloma macrophage and plays a role in mycobacterial immunity. It is conceivable that a more flexible monocyte precursor cell may develop into different macrophage types in the granuloma. However, deciphering development of foam cells in the granuloma seems beyond the scope of this manuscript

2.5. *The claim that "monocyte progenitors are specific effector cells in mycobacterial immunity" (line 38) seems unwarranted. And so does "This spurs the hypothesis that iMoP are involved in the defense against mycobacteria" (lines 354-355). In the first place, the authors do not test the antimycobacterial capacity of cMoPs, and secondly, this hypothesis clashes with their previous findings showing that MGCs are less microbicidal against*

mycobacteria than their macrophage counterparts. Furthermore, addressing the microbicidal capacity of these myeloid populations seems like a distraction from the main goal of this manuscript of understanding MGC development.

We appreciate the reviewer's thoughtful comment. We agree that concentrating on MGC development improves the focus of the paper. Moreover, effector cell is a controversial term. To accommodate this, we have, first, changed the title (now "Monocyte progenitors give rise to multinucleated giant cells") and second altered the quoted sentences in the summary and in line 401. We analysed in a killing assay the capacity of cMoP and iMoP to kill *M. bovis* BCG compared to MC (**new fig. S5F**). Referred to the cell number we found less bacteria per cell in the precursor subsets, but they also failed to completely kill the bacteria. We wanted to highlight that the precursor cells can develop into MGC which are involved in the immune response to mycobacteria. Overall, we fully agree that it remains speculative at this stage whether MGC primarily regulate the tissue reaction or are truly antimycobacterial as such.

Minor concerns:

2.6. *The title of the manuscript does not accurately reflect the primary focus of the paper. The manuscript seems more about MGC development than about the antimycobacterial capacity of MGCs*

In accordance with both of the reviewers' requests we have modified the title to

Monocyte progenitors give rise to multinucleated giant cells

2.7. *The introduction and discussion sections talk about the co-evolution of pathogenic mycobacteria and humans and allude to the possibility that MGCs might serve as a growth permissive niche for Mtb and may therefore contribute to latent TB. These statements detract from the main thrust of the paper: understanding how MGCs develop. Given that MGCs are observed in active tuberculosis and soon after mycobacterial stimulation in vitro, it might be more appropriate to focus the text on MGC development.*

We very much appreciate the reviewer's help with keeping the focus. Indeed, our manuscript centers around the origin of MGC. Nevertheless, we believe that deciphering granuloma biology at the example of specialized macrophage development sheds some light on coevolution between humans and mycobacteria. Our extended metabolic characterization with sterol analysis revealed accumulation of specific cholesterol precursors in developing

MGC. Moreover, we identified the regulation of cholesterol and lipid metabolism to be essential for cellular transformation. As it is reported that mycobacteria need cholesterol as a carbon source, a high availability of lipids might explain, why MGC are attractive host cells, thereby potentially contributing to mycobacterial persistence. Besides, it was reported that granulomas with their specific cellular composition, including MGC, not only exist in active disease, but also in latent stages of infection, and that they undergo a characteristic development with respect to e.g. vascularization, cell content and caseous necrosis (Russell et al. 2010. *Science* 328(5980): 852-6).

REVIEWER COMMENTS

Reviewer #1 (Remarks to the Author):

The authors have addressed most of my previous concerns and the manuscript has been significantly improved. This work constitutes a significant advance in our understanding of MGC formation in the context of TB infection.

Overall, the current version of the *in vitro* experimental approaches provides clear evidence that cMoP is more prone to give rise MGC, unlike MC or MDP (fig 1). Also, the authors demonstrate that cholesterol metabolism is link to MGC formation (fig 4). cMoP cells are known to be precursors of MC. Unlike MC generated under physiological conditions (fig 1E), those MC/M ϕ derived from cMoP under infectious settings differentiate into MGC (fig 2 and 3).

In addition, the current *in vivo* experimental approaches are quite convincing that iMoP cells, which are derived from cMoP under infection conditions (fig 5) and can give rise to circulating MC (fig. 6). Moreover, these iMoP cells migrate to spleen and liver granulomas in a mouse model of disseminated TB (fig 7). Despite the improvement above, the major critical issue is still not addressed by the authors: do iMoP truly become MGC in the *in vivo* setting once they reach the granuloma tissue? Yet, considering the experimental difficulties associated to this aim, I think that the fact that iMoP-derived-cells accumulated at the sites of infection, and particularly within tuberculous tissue structures, is very promising and constitutes an important finding introduced in this new version of the manuscript.

Although the authors have addressed most of my scientific concerns, it would be desirable that the authors focus on making the manuscript text a more readable and easier to follow. For example, it would be desirable that figures — especially supplementary ones — to be enumerated following the order in which they are referred to throughout the manuscript text.

Finally, I suggest some minor changes and corrections in the manuscript:

- Lines 130-133. The new sentence concerning the analysis of binucleated cells is not comprehensive at all. In order to help the reader to understand why binucleated cells are not informative, I suggest introducing the notion that binucleated cells may comprise a snapshot of cells, which are under proliferation instead of becoming MGC.
- Line 155. When the authors conclude that TLR2 is not a limiting factor in MGC formation by MC cells, it sounds too categorical taken into consideration that this conclusion is merely based on mRNA expression.
- Line 192: monocytes should be changed to MC.
- It is not clear why ECAR is shown in the Mito Stress test (figure S3A), instead of showing the OCR results. What does the acidification of the media exactly mean when applying the sequence of drugs of the Mito Stress test? If results do not provide fundamental data to sustain the switch towards glycolysis, I recommend removing this panel, keeping only the information associated to panel B.
- Line 367: please clarify that MC-21 is an anti-CCR2 antibody. Also fig. S4D should be fig. S5D.
- Line 370: Tb should be M. tb
- Line 371: fig. S4E should be fig. S5E.
- Lines 371 – 376 concerning the possible reason by which MGC are barely detectable in lungs of infected mice, the data is quite preliminary and speculative. MGC are missing in lung parenchyma, thus BAL milieu may not necessarily be involved. Besides, would you suggest that BAL milieu differs between humans and mice in order to explain the lack of MGC in the murine model? Further studies are needed to present this data in the results section, and specially when these results do not further strengthen the main finding of this manuscript. Based on it, I suggest removing this paragraph and adapt the discussion accordingly.
- The manuscript is slightly biased towards demonstrating that cMoP and iMoP display antimycobacterial effector properties. Concerning the connection between MGC and their postulated precursors, cMoP, with the foamy phenotype, the authors may consider the possibility that these cells may bear dormant bacteria. In this sense, Tornack et al reported that human peripheral blood long-term pluripotent hematopoietic stem cells harvested from IGRA-positive asymptomatic individuals contained M. tb DNA, whereas those from IGRA-negative individuals did not (PMID: 28046053). Perhaps, it could provide an alternative interpretation of the role of these MGC precursors to be included in the discussion section.

Sincerely,
Dr. Luciana Balboa

Reviewer #2 (Remarks to the Author):

I appreciate the authors' considerable efforts to address reviewer criticisms and improve the manuscript. Nonetheless, there are still some critical issues raised during the initial review that remain unclear.

In order to determine whether or not circulating monocyte progenitors can participate in mycobacterial granulomas, the authors infected mice with BCG intravenously, adoptively-transferred FACS-purified GFP-expressing monocytes, and quantified the proportion of GFP+ cells associated with granulomas in the liver and spleen. While this experiment nicely demonstrates that circulating monocyte progenitors can be recruited to granulomas, the analyses presented do not address whether any of the transferred cells found in granulomas have become multinucleated giant cells (MGC).

In Figure 7H-J, the authors show Hoechst staining. Couldn't this be used to determine what percentage of the GFP+ cells have become multinucleated? Furthermore, couldn't gH2AX staining be used to link monocyte precursor recruitment to granulomas and the DNA damage response implicated in MGC formation? In a previous paper, the authors used gH2AX staining to provide *in vivo* evidence that the DNA damage they had described *in vitro* was also unfolding *in vivo* (Herrtwich, et al. *Cell*. 2016). Experiments along these lines would strengthen the claim from the present manuscript that circulating monocyte progenitors are a physiological source of MGC.

The CCR2 blockade experiments show a depletion of common myeloid progenitors (cMoP) and induced monocyte precursors (iMoP) in the bone marrow, in addition to the expected decrease of iMoP in the blood. The loss of cMoP and iMoP in the bone marrow suggests the CCR2 antibody MC-21 is actually depleting CCR2-expressing cells rather than impeding their egress from the bone marrow, an issue previously articulated by Prinz and colleagues (Mildner, et al. *Nat. Neuroscience* 2007). If this antibody were simply blocking CCR2 signaling, one would expect an increase in cMoP and iMoP frequencies in the bone marrow due to sequestration of these cells in the tissue, as was previously shown for CCR2^{-/-} monocytes (Serbina and Pamer. *Nat. Immunol.* 2006). In light of this information, the CCR2 antibody experiments should be reinterpreted and the text in lines 362-364 and 494-496 should be modified accordingly. To reiterate, the experiments shown do not test mobilization of the cells in question; fewer iMoP were most likely found in the blood because they were depleted there and in the BM rather than due to inhibiting their migration.

Another outstanding question pertaining to the CCR2 experiments is whether or not the antibody treatment impaired MGC formation *in vivo*. The authors provide some reasons for as to why testing this *in vivo* is fraught with caveats, invoking roles of CCR2 in other cell types and cellular processes, but go on to show that in their experiments, antibody treatment did not grossly affect bacterial burdens in the lung or spleen, findings consistent with the observation that CCR2^{-/-} mice are able to control low dose aerosol Mtb infection (Scott and Flynn. *Infec. and Immun.* 2002). To me, this means that the authors should at least try to address the questions I posed in the initial review. If a CCR2-expressing cell is the primary MGC precursor, then the antibody treatment should reduce their abundance in the granuloma. This is an important issue as other myeloid cells in the granuloma, e.g., tissue resident macrophages or CCR2-negative monocytes might potentially serve as alternative sources of MGCs *in vivo*.

The physiological significance of this interesting study could be substantially improved by addressing the issues highlighted above.

Point-by-point response to the reviewers' comments:

Reviewer #1:

The authors have addressed most of my previous concerns and the manuscript has been significantly improved. This work constitutes a significant advance in our understanding of MGC formation in the context of TB infection.

Overall, the current version of the in vitro experimental approaches provides clear evidence that cMoP is more prone to give rise MGC, unlike MC or MDP (fig 1). Also, the authors demonstrate that cholesterol metabolism is link to MGC formation (fig 4). cMoP cells are known to be precursors of MC. Unlike MC generated under physiological conditions (fig 1E), those MC/Mφ derived from cMoP under infectious settings differentiate into MGC (fig 2 and 3). In addition, the current in vivo experimental approaches are quite convincing that iMoP cells, which are derived from cMoP under infection conditions (fig 5) and can give rise to circulating MC (fig. 6). Moreover, these iMoP cells migrate to spleen and liver granulomas in a mouse model of disseminated TB (fig 7).

1.1 Despite the improvement above, the major critical issue is still not addressed by the authors: do iMoP truly become MGC in the in vivo setting once they reach the granuloma tissue? Yet, considering the experimental difficulties associated to this aim, I think that the fact that iMoP-derived-cells accumulated at the sites of infection, and particularly within tuberculous tissue structures, is very promising and constitutes an important finding introduced in this new version of the manuscript.

We appreciate that the depiction of MGC transformation in the granuloma at the single cell level *in vivo* is a – experimentally highly challenging – milestone in understanding granuloma biology. In the previous version of the manuscript we had already provided evidence for transformation of transferred progenitors into specialized granuloma macrophages. Here, we have tackled the substantial inherent experimental challenges on collecting evidence for actual MGC transformation of monocyte progenitors *in vivo*. To this end we made use of IL-13^{tg} mice with pulmonary *M.tb* infection. These mice not only form granulomas with central necrosis, but also multinucleated giant cells (MGC), which distinguishes them from most Tb mouse models. Six weeks after *M.tb* aerosol infection of IL-13^{tg} mice, we intratracheally transferred CD115⁺ Ly6C⁺ CD11b⁻ cells, including cMoP and iMoP, from the bone marrow of *β-Actin gfp^{+/+}* mice. Three weeks after cell transfer, i.e. 9 weeks after infection, we could indeed identify individual transferred cells with three nuclei by microscopy of infected lungs,

in line with our definition of MGC. Additionally, staining of nitric oxide synthase (iNOS), which is a suitable cell-intrinsic MGC marker¹ revealed iNOS-positive transferred cells (NEW **figs. 8D and 8E, lines 510-531**). Together, these data provide further evidence on the *in vivo* formation of progenitor-derived MGC. Of note, the contribution of a T helper 2 inflammatory environment for progenitor cells to acquire MGC properties remains to be established.

1.2 Although the authors have addressed most of my scientific concerns, it would be desirable that the authors focus on making the manuscript text a more readable and easier to follow. For example, it would be desirable that figures — especially supplementary ones — to be enumerated following the order in which they are referred to throughout the manuscript text.

We fully agree with the reviewer that a coherent order of text passages and figures is important for the understanding of the manuscript. We have therefore revised the sequence of the figures.

Minor concerns

1.3 Lines 130-133. The new sentence concerning the analysis of binucleated cells is not comprehensive at all. In order to help the reader to understand why binucleated cells are not informative, I suggest introducing the notion that binucleated cells may comprise a snapshot of cells, which are under proliferation instead of becoming MGC.

We thank the reviewer for these considerations. Accordingly, we have introduced binucleated cells as potentially proliferating cells and rephrased the respective paragraph (**lines 153-161**).

1.4 Line 155. When the authors conclude that TLR2 is not a limiting factor in MGC formation by MC cells, it sounds too categorical taken into consideration that this conclusion is merely based on mRNA expression. Line 192: monocytes should be changed to MC.

We have revised this categorical statement (**line 183**) and changed monocytes to MC (**line 219**).

1.5 It is not clear why ECAR is shown in the Mito Stress test (figure S3A), instead of showing the OCR results. What does the acidification of the media exactly mean when applying the sequence of drugs of the Mito Stress test? If results do not provide fundamental data to

sustain the switch towards glycolysis, I recommend removing this panel, keeping only the information associated to panel B.

We agree with the reviewer that the Mito Stress Test may be difficult to be interpreted. Thus, we have removed the respective data and have focused on the glycolysis stress test (former fig. S3B, **now S4A**).

1.6

Line 367: please clarify that MC-21 is an anti-CCR2 antibody. Fig. S4D should be fig. S5D.

Line 370: Tb should be M.tb

Line 371: fig. S4E should be fig. S5E.

We have corrected accordingly.

1.7 *Lines 371 – 376 concerning the possible reason by which MGC are barely detectable in lungs of infected mice, the data is quite preliminary and speculative. MGC are missing in lung parenchyma, thus BAL milieu may not necessarily be involved. Besides, would you suggest that BAL milieu differs between humans and mice in order to explain the lack of MGC in the murine model? Further studies are needed to present this data in the results section, and specially when these results do not further strengthen the main finding of this manuscript. Based on it, I suggest removing this paragraph and adapt the discussion accordingly.*

We thank the reviewer for these important considerations. Indeed, the data set with respect to the influence of the BAL composition and species-specific differences is somewhat preliminary. Therefore, we have removed the data and have adjusted the discussion in compliance with the reviewer's suggestions.

1.8 *The manuscript is slightly biased towards demonstrating that cMoP and iMoP display antimycobacterial effector properties. Concerning the connection between MGC and their postulated precursors, cMoP, with the foamy phenotype, the authors may consider the possibility that these cells may bear dormant bacteria. In this sense, Tornack et al reported that human peripheral blood long-term pluripotent hematopoietic stem cells harvested from IGRA-positive asymptomatic individuals contained M.tb DNA, whereas those from IGRA-negative individuals did not (PMID: 28046053). Perhaps, it could provide an alternative interpretation of the role of these MGC precursors to be included in the discussion section.*

We have incorporated this important alternative hypothesis in our discussion (**lines 641-645**). Due to the lipid transformation, which we observed in progenitor cells upon TLR2 stimulation, they might provide a suitable environment to host and carry dormant mycobacteria.

Reviewer #2

I appreciate the authors' considerable efforts to address reviewer criticisms and improve the manuscript. Nonetheless, there are still some critical issues raised during the initial review that remain unclear.

2.1. *In order to determine whether or not circulating monocyte progenitors can participate in mycobacterial granulomas, the authors infected mice with BCG intravenously, adoptively-transferred FACS-purified GFP-expressing monocytes, and quantified the proportion of GFP+ cells associated with granulomas in the liver and spleen. While this experiment nicely demonstrates that circulating monocyte progenitors can be recruited to granulomas, the analyses presented do not address whether any of the transferred cells found in granulomas have become multinucleated giant cells (MGC).*

In Figure 7H-J, the authors show Hoechst staining. Couldn't this be used to determine what percentage of the GFP+ cells have become multinucleated? Furthermore, couldn't gH2AX staining be used to link monocyte precursor recruitment to granulomas and the DNA damage response implicated in MGC formation? In a previous paper, the authors used gH2AX staining to provide in vivo evidence that the DNA damage they had described in vitro was also unfolding in vivo (Herrtwich, et al. Cell. 2016). Experiments along these lines would strengthen the claim from the present manuscript that circulating monocyte progenitors are a physiological source of MGC.

We thank the reviewer for these considerations. To further strengthen the hypothesis that progenitor cells participate in MGC formation *in vivo*, we have, as outlined above, performed an intratracheal adoptive progenitor transfer into *M.tb* infected IL-13^{tg} mice. These mice develop granulomas with histological structures similar to those found in tuberculosis patients, including central necrosis and multinucleated giant cells after infection with *M.tb* via aerosol in contrast to most Tb mouse models². Intratracheal transfer of CD115⁺ Ly6C⁺ CD11b⁻ cells isolated from β -Actin *gfp*^{+/+} mice was performed six weeks p.i.. Detailed microscopic analysis nine weeks after infection revealed individual GFP⁺ cells with 3 nuclei in the lungs of IL-13^{tg} mice, in line with our definition as MGC.

To define the number of nuclei we made use of Hoechst staining as suggested by the reviewer. Additionally, we performed staining of the inducible nitric oxide synthase (iNOS)

which is a characteristic cell intrinsic marker for the MGC transformation¹ and found individual transferred cells to be positive for iNOS. With this, we have added a completely new data set substantially supporting the hypothesis that progenitor cells contribute to MGC and granuloma formation in mycobacterial infections (**figs. 8D and 8E, lines 510-531**).

2.2. *The CCR2 blockade experiments show a depletion of common myeloid progenitors (cMoP) and induced monocyte precursors (iMoP) in the bone marrow, in addition to the expected decrease of iMoP in the blood. The loss of cMoP and iMoP in the bone marrow suggests the CCR2 antibody MC-21 is actually depleting CCR2-expressing cells rather than impeding their egress from the bone marrow, an issue previously articulated by Prinz and colleagues (Mildner, et al. Nat. Neuroscience 2007). If this antibody were simply blocking CCR2 signaling, one would expect an increase in cMoP and iMoP frequencies in the bone marrow due to sequestration of these cells in the tissue, as was previously shown for CCR2^{-/-} monocytes (Serbina and Pamer. Nat. Immunol. 2006). In light of this information, the CCR2 antibody experiments should be reinterpreted and the text in lines 362-364 and 494-496 should be modified accordingly. To reiterate, the experiments shown do not test mobilization of the cells in question; fewer iMoP were most likely found in the blood because they were depleted there and in the BM rather than due to inhibiting their migration.*

We thank the reviewer for these considerations. We have revised the interpretation of the effect of the anti-CCR2-antibody MC-21. Additionally, we have performed intravenous *M. bovis* BCG infection of CCR2-deficient mice. In contrast to application of the MC-21 antibody, we found an accumulation of progenitor cells in the bone marrow of these mice 18 days post infection. At the same time, we observed only a minimal increase in CD115⁺ CD11b⁻ Ly6C⁺ cells in the spleens compared to infected wildtype controls. These data further sustain the hypothesis that the recruitment of progenitor cells is CCR2-dependent (**NEW figs. 7E and 7F, lines 483-494**).

2.3. *Another outstanding question pertaining to the CCR2 experiments is whether or not the antibody treatment impaired MGC formation in vivo. The authors provide some reasons for as to why testing this in vivo is fraught with caveats, invoking roles of CCR2 in other cell types and cellular processes, but go on to show that in their experiments, antibody treatment did not grossly affect bacterial burdens in the lung or spleen, findings consistent with the observation that CCR2^{-/-} mice are able to control low dose aerosol *Mtb* infection (Scott and Flynn. Infec. and Immun. 2002). To me, this means that the authors should at least try to address the questions I posed in the initial review. If a CCR2-expressing cell is the primary MGC precursor, then the antibody treatment should reduce their abundance in the*

granuloma. This is an important issue as other myeloid cells in the granuloma, e.g., tissue resident macrophages or CCR2-negative monocytes might potentially serve as alternative sources of MGCs in vivo.

We agree with the reviewer that it is important to study the role of progenitor cells in the context of CCR2-deficiency. Unfortunately, the spontaneous formation of neutralizing antibodies against MC-21 antibodies in treated mice does not allow for depletion experiments exceeding one week to study MGC formation *in vivo*. Instead, we have analyzed pulmonary *M.tb* infected CCR2-deficient mice 60 to 300 days post infection with respect to MGC formation. The relative immune cell composition was largely similar in both wildtype and CCR2^{-/-} mice including MGC with high lipid content and foam cell formation after 100 and 300 days of infection (NEW **figs. S5E, S5F, S5G**). Of note, isolated progenitor cells from wildtype and CCR2^{-/-} have a mostly similar capacity to form MGC *in vitro* (NEW **figs. S5H and S5I**), (**lines 394-415**). This data together with the BCG-infection of CCR2^{-/-} mice not only demonstrates that MGC formation is per se possible in CCR2-deficient mice, but also that CCR2-deficient progenitor cells possess the essential requirements to form MGC. Although we could show that the egress of progenitor cells from the bone marrow is CCR2 dependent, leading to a reduced availability of these cells, the remaining progenitor cells, which we identified for example in the spleen (NEW **fig. 7E**), might fill the “MGC progenitor niche” during mycobacterial infections. We have elaborated on these issues in the discussion.

1. Gharun, K. *et al.* Mycobacteria exploit nitric oxide-induced transformation of macrophages into permissive giant cells. *EMBO Rep.* **18**, 2144–2159 (2017).
2. Heitmann, L. *et al.* The IL-13/IL-4R α axis is involved in tuberculosis-associated pathology. *J Pathol* **234**, 338–350 (2014).

REVIEWERS' COMMENTS

Reviewer #1 (Remarks to the Author):

The authors have adequately addressed/clarified my previous critiques of this work. The finding of MGC derived from bone marrow progenitors in lung granulomas from M.tb infected IL-13tg mice sustancially improves the physiological relevance of the study.

Reviewer #2 (Remarks to the Author):

The authors have addressed my concerns satisfactorily.

Point-by-point response to the reviewers' comments:

Reviewer #1 (Remarks to the Author):

The authors have adequately addressed/clarified my previous critiques of this work. The finding of MGC derived from bone marrow progenitors in lung granulomas from M.tb infected IL-13tg mice substantially improves the physiological relevance of the study.

Reviewer #2 (Remarks to the Author):

The authors have addressed my concerns satisfactorily.

We thank the reviewers for all of their efforts and important suggestions during the review process, which have substantially improved the manuscript.